# Interface controlled thermal resistances of ultra-thin chalcogenide-based phase change memory devices

Kiumars Aryana[1], John T. Gaskins [1], Joyeeta Nag[2], Derek A. Stewart [2], Zhaoqiang Bai[2], Saikat Mukhopadhyay[3], John C. Read[2], David H. Olson[1], Eric R. Hoglund [4], James M. Howe[4], Ashutosh Giri[5], Michael K. Grobis[2] & Patrick E. Hopkins [1,4,6✉]

Phase change memory (PCM) is a rapidly growing technology that not only offers advancements in storage-class memories but also enables in-memory data processing to overcome the von Neumann bottleneck. In PCMs, data storage is driven by thermal excitation. However, there is limited research regarding PCM thermal properties at length scales close to the memory cell dimensions. Our work presents a new paradigm to manage thermal transport in memory cells by manipulating the interfacial thermal resistance between the phase change unit and the electrodes without incorporating additional insulating layers. Experimental measurements show a substantial change in interfacial thermal resistance as GST transitions from cubic to hexagonal crystal structure, resulting in a factor of 4 reduction in the effective thermal conductivity. Simulations reveal that interfacial resistance between PCM and its adjacent layer can reduce the reset current for 20 and 120 nm diameter devices by up to ~ 40% and ~ 50%, respectively. These thermal insights present a new opportunity to reduce power and operating currents in PCMs.

[1] Department of Mechanical and Aerospace Engineering, University of Virginia, Charlottesville, VA 22904, USA. [2] Western Digital Corporation, San Jose, CA 95119, USA. [3] NRC Research Associate at Naval Research Laboratory, Washington, DC 20375, USA. [4] Department of Materials Science and Engineering, University of Virginia, Charlottesville, VA 22904, USA. [5] Department of Mechanical, Industrial and Systems Engineering, University of Rhode Island, Kingston, RI 02881, USA. [6] Department of Physics, University of Virginia, Charlottesville, VA 22904, USA. ✉email: peh4v@virginia.edu

The growing demands for higher capacity memory devices and burgeoning data-intensive applications, such as artificial intelligence, have intensified efforts to beat the von Neumann computing bottleneck that separates processing from the storage unit. A promising alternative for transistor-based non-volatile memory devices is an emerging technology known as phase change memory (PCM), which offers prospective gains in speed, device lifetime, and storage capacity, as well as in-memory storage and computing capabilities[1,2]. The most widely used phase change material, germanium antimony telluride (GST), possesses a high electrical resistivity contrast between its amorphous and crystalline structure, as well as sub-nanosecond switching times[3,4]. This class of phase change materials can quickly switch phase between amorphous and crystalline states upon controlled thermal excitation. In PCMs, the transition from amorphous to crystalline and crystalline to amorphous are commonly referred to as set and reset, respectively. In devices utilizing phase change units, thermal transport plays a pivotal role as it dictates the efficiency of the set/reset process and overall power consumption.

One of the major limitations in PCM devices is their high operating current, leading to excessive power consumption[5]. In order to mitigate thermal leakage during programming, Kim et al.[6] used a thermal barrier (2–20 nm of $C_{60}$) to insulate the GST from directly contacting the electrode, showing up to a factor of four reduction in their reset current ($I_{reset}$). Although a lower power consumption in their device architecture offered performance gains, the relatively large thickness of the thermal barrier introduced additional resistance, decreased bit density, and provided an additional source of degradation for the PCM over time. Later, Ahn et al.[7] proposed a much thinner insulating layer by using a single sheet of graphene (thickness <1 nm) as a thermal barrier to isolate the heat inside the PCM cell and showed that the $I_{reset}$ was reduced by 40% compared to the cells without a graphene barrier. More recently, superlattice phase change memories have received a great deal of attention due to their unique capabilities offering lower power consumption, faster programming rate, higher retention time, and lower noise and drift in electrical resistance[3,8–10]. Although earlier superlattice PCMs consisted of $GeTe/Sb_2Te_3$ alternating stacks, it was soon realized that this configuration tends to intermix and transform into bulk GST at high annealing temperatures[11]. Nonetheless, the idea of superlattice PCMs inspired researchers to look for alternative material configurations. Very recently, Shen et al.[8] and Ding et al.[9] showed that superlattice PCMs with $TiTe_2/Sb_2Te_3$ layers have superior properties compared to bulk GST. Despite the fact that in superlattice PCMs the interface is an integral component in the performance of these devices, its effect on the overall thermal transport is heretofore unknown and unstudied. With all these previous works in mind, we are prompted to experimentally investigate the effect of interfacial thermal resistance on the performance of PCM devices. The selected materials for this study are among those that are widely used in PCM devices: $Ge_2Sb_2Te_4$ (GST) as a phase change unit, tungsten (W) as an electrode, and silicon dioxide ($SiO_2$) and silicon nitride ($SiN_x$) as the insulating separators for confining heat and current within the cell. Our work focuses on identifying the critical parameters that influence thermal transport as the length scale of the phase change unit approaches that of energy carriers' mean free paths. We assess the effect of GST film thickness on thermal transport across various phase transitions and determine the minimum thickness before which thermal transport transitions into a ballistic regime. A pictorial representation of the configuration of layers used in this study is given in Fig. 1a along with the corresponding transmission electron microscopy (TEM) images for amorphous (a-GST) and hexagonal (h-GST) phases.

To date, the majority of studies investigating thermal transport in GST were performed on layers with thicknesses on the order of 200 nm[12–15]. However, as Xiong et al.[16] demonstrated, in order to decrease power consumption and further the economic benefits of PCM memory devices, the thickness of GST layers should be on the order of 10 nm. In this respect, Kim et al.[17] devised an operational PCM device with cell dimensions as small as 7.5 nm × 17 nm. In general, as the length scale of materials and interconnects in PCM components shrink to dimensions less than energy carrier mean free paths, a number of additional mechanisms, such as electron tunneling[18–20] and thermal boundary resistances (TBRs)[21–23], may impact the performance of these devices[24]. In this paper, we present evidence of ballistic transport of energy carriers across the electrodes in a confined cell geometry as the characteristic length of the device is decreased to less than the mean free paths of the electrodes carriers. To demonstrate this, we show that in tungsten electrodes there is a lower limit for the thickness of GST in memory cells before thermal transport transitions from a diffusive to a ballistic regime. Thus, our work uses this knowledge of carrier dynamics to experimentally identify an optimal thickness of phase change material based on a balance of thermal conductivity and crystallographic-phase-dependent thermal boundary conductances (TBCs) in order to improve memory device performance.

Here, in contrast with previous studies that were primarily focused on introducing additional layers between the electrode and GST to confine heat in the memory cell, we focus on the interfacial thermal resistance and thermal properties of the layers in contact with GST. We show that, by intentionally engineering the phase and thickness of the phase change unit, the overall thermal resistance can be substantially increased, causing decreases in requirements for set/reset currents, without incorporating additional layers as a thermal barrier. Although the results presented here are for commonly used materials in PCMs such as GST and W, we demonstrate that, through manipulation of the interfacial resistance between the phase change unit and the adjacent layer, the predicted reset current can be reduced by up to 40% and 50% for devices with lateral size of 20 and 120 nm in diameters, respectively. Our work highlights the importance of engineering interfaces to allow for devices with increased performance.

## Results

We measure the thermal transport properties of the GST thin films, deposited via magnetron sputtering, using time-domain thermoreflectance (TDTR) in a two-tint configuration[25]. The surface of the samples are coated with an 80-nm ruthenium transducer. The thermal model, which relates the thermoreflectivity of the transducer to the thermal properties of the underlying layers, requires knowledge of the volumetric heat capacity, film thickness, and thermal conductivity of each layer. The volumetric heat capacity for Ru, a-GST, h-GST, and the Si substrate are assumed to be 2.96, 1.3, 1.4, and 1.64 MJ $m^{-3}$ $K^{-1}$, respectively[26,27]. The thermal conductivity of the Ru layer is determined to be 54 W $m^{-1}$ $K^{-1}$ via four-point probe measurements and the thickness of each layer is confirmed via TEM.

**Room temperature thermal properties of a-GST.** We first investigate room temperature thermal properties in order to show that the effect of materials adjacent to GST, spacers, on thermal conductance is only appreciable for a-GST thicknesses less than 10 nm. For films as thin as these, it is often instructive to idealize material stacks as a series of thermal resistors comprising the resistances at interfaces and the intrinsic resistance of the materials, similar to the schematic shown in Fig. 1c. The thermal

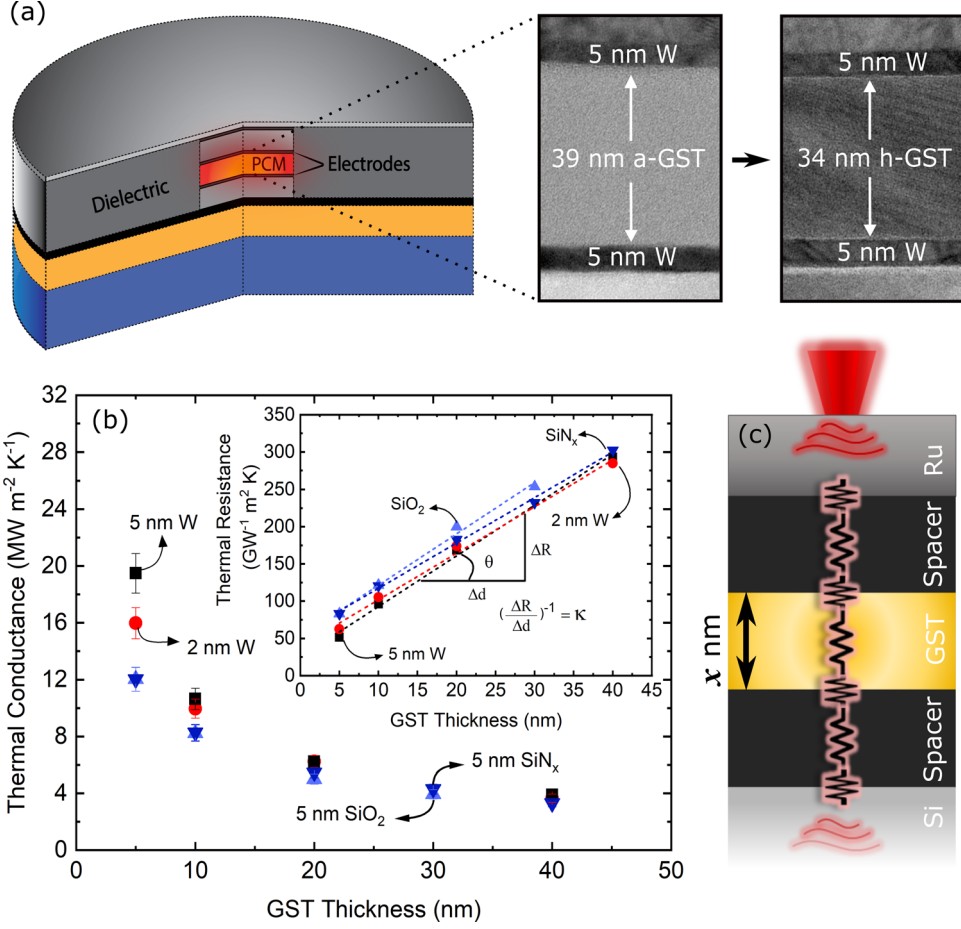

**Fig. 1 Thin GST films and contacts. a** Schematic of a confined phase change memory cell along with corresponding TEMs for a 40-nm a-GST and h-GST film sandwiched between 5 nm tungsten spacers, **b** thermal conductance across Ru/spacer/GST/spacer/Si for different spacer compositions as a function of GST thickness. The inset shows thermal resistance as a function of thickness where the inverse of the slope for the fitted line corresponds to the a-GST thermal conductivity. The average thermal conductivity estimated for a-GST is $0.15 \pm 0.02$ W m$^{-1}$ K$^{-1}$. The error bars are calculated based on 7% uncertainty in the Ru transducer thickness, and **c** a schematic of the thermal resistances in series for the multilayer configurations studied here.

resistance we measure, due to the thickness of the film stacks, is the total resistance between the Ru and Si. Using different thicknesses of a-GST, which varies the relative contribution of each resistor to the overall measured conductance, allows us to assess the relative contribution of each thermal resistance to the overall device thermal transport.

Figure 1b shows the thermal conductance between Ru and Si, including all intermediate layers (spacer/a-GST/spacer), as a function of a-GST thickness. The spacers we utilize are W, with thickness of 2 and 5 nm, amorphous $SiO_2$ (5 nm), and amorphous $SiN_x$ (5 nm), where the spacers' thicknesses are identical on either side of the a-GST. As the thickness of a-GST increases, the effect of the spacers on the overall thermal transport becomes negligible owing to the fact that the a-GST layer becomes the dominant resistor. Based on Fig. 1b, for thicknesses greater than ~10 nm, thermal conductance is largely governed by the a-GST layer regardless of the adjacent material, whereas for thicknesses less than 10 nm, the effect of TBC becomes appreciable. Note, for $SiN_x$ and $SiO_2$ spacers, their thermal conductances are similar and lower than that of W. This is expected as the thermal conductivities of $SiN_x$ and $SiO_2$ are similar and more than an order of magnitude lower than that of W[28]. However, it is important to note that the thermal conductance of the stack with the 5 nm W spacer is greater than that with 2 nm spacer. This is contrary to expectations when considering diffusive thermal transport processes, where thermal conductance decreases linearly with an increase in thickness of the

material. The observed reduction in thermal conductance for 2 nm W is attributed to the scattering of electrons and phonons at its boundaries. Similar TBC dependencies on the thickness of intermediate layer have been observed across Au/Ti/sapphire[29], Au/Cr/sapphire[30], and Au/Cu/sapphire[30] interfaces. The thermal conductivity of a-GST is determined from these measurements by fitting a linear regression to the slope of the measured thermal resistance as a function of thickness, depicted in the inset of Fig. 1b. The thermal conductivity of a-GST is determined to be $0.15 \pm 0.02$ W m$^{-1}$ K$^{-1}$, in good agreement with previously reported values[12,13,31].

**GST morphology at different phases**. In order to confirm phase transformation and the quality of the crystal structure associated with each phase, we perform TEM on the 40 and 160 nm GST with in situ heating (Figs. 2a–f). The transition from diffuse rings in the selected area diffraction (SADP) to sharp diffraction rings denotes the transformation from a-GST to polycrystalline cubic GST (c-GST), as shown in Figs. 2g and h, respectively. This is in agreement with previous results showing that GST transforms from an amorphous phase to a face-centered cubic lattice at ~155 °C[32–34]. The 160 nm GST film thickness was measured as 160, 152, and 149 nm at 25, 240, and 400 °C, respectively, and similarly the 40 nm GST film thickness as 38.7, 36.9, and 33.7 nm at 25, 240, and 400 °C. On average, the thickness of our GST films

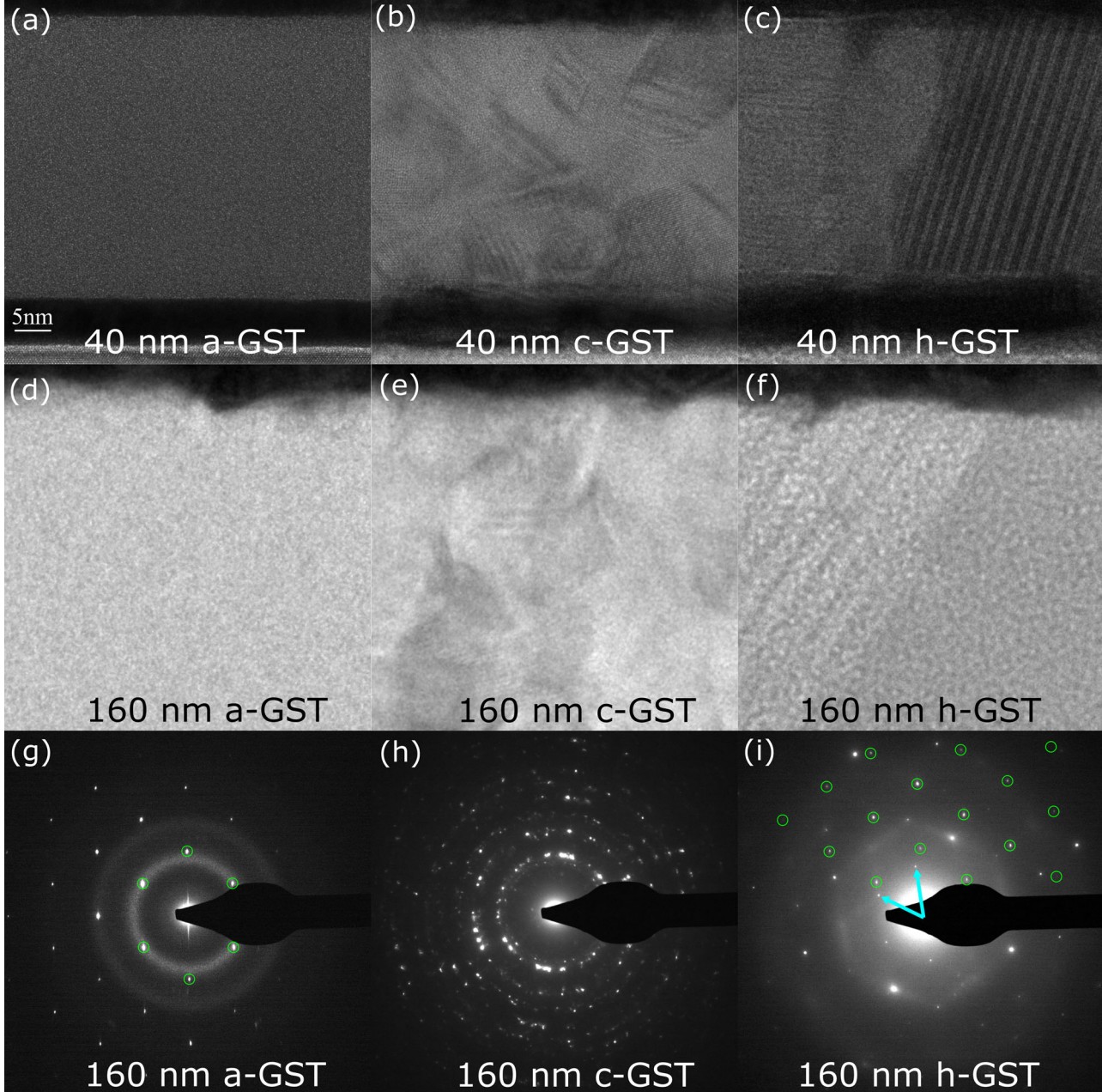

**Fig. 2 Transmission electron microscopy of different GST phases.** Bright-field images of the: **a–c** 40 nm and **d–f** 160 nm GST thin films at: **a, d** 25 °C, **b, e** 240 °C and **c, f** 400 °C, showing a sequence of phase transformations from amorphous to cubic to hexagonal GST, respectively. Selected area diffraction patterns (SADP) of the 160 nm film at: **g** 25, **h** 240, and **i** 400 °C, reflecting the phase transformations and microstructures observed in the BF images. In the inner ring of **g** and the top half of **i**, the green circles indicate Bragg spots in the [110] zone axis of the Si substrate, and blue arrows in **i** show {100} primitive reciprocal-lattice vectors of h-GST in a [001] zone axis. Diffuse streaks in **i** extending through the {2$\bar{1}$0} Bragg spots are due to the transnational shear faults seen in **c** and **f**.

change by ~5% and ~6%, at the transition from amorphous to cubic and cubic to hexagonal, respectively, which are comparable to the values of 6.5% and 8.2% reported elsewhere[35]. The grain size for c-GST ranges from 10 to 20 nm in both films. When the films transform to the hexagonal phase, there are highly faulted grains that span the thickness of the films, as shown in Fig. 2c. A few differences in the lateral (i.e., in-plane) grain size were observed between the 40 and 160 nm films. In the 160 nm film, the lateral size of the smaller grains is approximately 50 nm, whereas in the 40 nm film, the size varied from 50 to 100 nm. Large faulted grains in the 160 nm film are ~100–200 nm wide, but only about 100 nm wide in the 40 nm film. The cross-plane

dimension of the grains in the 40 nm film are often that of the film thickness (i.e., 40 nm), while there is a range in the 160 nm film. The SADP from GST at 400 °C (Fig. 2i) displays a single GST [001] zone axis and the [110] zone axis of the Si substrate, due to the much larger grain size compared to 240 °C (Fig. 2h). Diffuse streaks emanate from the {2$\bar{1}$0} Bragg spots as a result of the faulted grain. Highly faulted structures have been observed in a variety of chalcogenides including Ge–Sb–Te compounds[36–41]. The large, faulted grains grew laterally by growth ledges that often nucleate at the W/GST interface. The growth ledges then propagate along the h-GST/c-GST interface, consuming smaller grains as they move. Crystalline GST is composed of van der

Waals coupled building blocks, each of which contains five Te layers separated by either a Ge or an Sb anion layer[36,37,42,43]. By shifting each nine-layer building block by a partial lattice vector, the c-GST becomes h-GST and vice versa. Recent literature suggests that the weak bonding between the sesqui-chalcogenides building blocks, such as $Sb_2Te_3$, significantly exceeds those of van der Waals forces and, therefore, possesses more of a metavalent bond nature[40,44]. As a result of the weak bonding between the blocks, there is a low-energy barrier to passing partial dislocations that transform the lattice and cause faults. Faulted grains in the 160 nm sample were primarily at an angle to the film normal as seen in Fig. 2f. The same was observed for the 40 nm film, such as on the right side of Fig. 2c in addition to grains whose building blocks/faults were parallel to the film normal, as on the left side of Fig. 2c.

**Elevated temperature thermal properties of GST**. Above, we showed that the effects of room temperature TBC values on overall device thermal resistance are only appreciable for a-GST when the thickness is less than 10 nm. However, as the a-GST film changes phase, its intrinsic thermal conductivity increases by almost an order of magnitude. This implies that thermal transport in the crystalline phase should be more dramatically affected by the TBC than in the amorphous phase. Therefore, it is crucial to understand the effects of thermal transport across W/GST interfaces as GST undergoes phase transition. TDTR measurements are taken as a function of temperature using a resistive heating stage that allows us to measure the thermal conductivity of GST and the TBC at the GST/W interface from room temperature up to 400 °C, thereby, capturing the thermal properties of GST in all of phases (i.e., amorphous, cubic and hexagonal). Although cross-plane electrical resistivity measurements for GST films are beyond the scope of this paper, in Supplementary Note 1 we use available electrical resistivity data in the literature to differentiate the contributions of electrons and phonons to the total thermal conductivity of GST.

Figure 3a shows the thermal conductivity of 40-nm and 160-nm-thick a-GST layers that are heated under nominally identical conditions across various temperatures. In this figure, the solid symbols correspond to the thermal conductivity of a-GST when heated from room temperature up to 400 °C, whereas the hollow symbols correspond to the thermal conductivity of h-GST when cooled down from 400 °C to room temperature. The solid circles denoting the 160 nm film in Fig. 3a show a clear transition from a-GST to c-GST, and c-GST to h-GST at approximately 140 and 340 °C, respectively (see Supplementary Movies 1 and 2), in good agreement with reported literature values[12,45,46]. The enhancement of thermal conductivity in the crystalline phase is attributed to the dissolution of disordered vacancy clusters and increasing order in the crystalline phase[47,48]. After the sample reaches 400 °C and the GST is fully transformed into the hexagonal phase, its thermal conductivity is measured as the sample is cooled down to room temperature, shown as hollow circles in Fig. 3a. The thermal conductivity of the 160 nm h-GST decreases slightly over temperature as a result of reduced electronic contribution to the thermal conductivity[47].

On the other hand, for the 40-nm-thick GST, the measurement of intrinsic thermal conductivity in the crystalline phase is increasingly difficult as the effects of interfacial thermal resistance interfere with thermal conductivity measurements as opposed to the 160 nm case (see Supplementary Note 2). For this reason, we report the effective thermal conductivity ($k_{eff} = G_{Ru/W/GST/W/Si} \times d_{GST}$), depicted as solid diamonds, which incorporates both the effects of the intrinsic thermal conductivity of GST and the associated TBCs.

The effective thermal conductivity for the 40-nm-thick GST sample follows a similar trend to that of 160 nm film up to 300 °C, except for the slight upward shift in crystallization temperature to 150 °C. The agreement of thermal conductivity up to 300 °C between the two thicknesses is due to negligible effect of TBC on thermal transport in the amorphous and cubic phases. However, upon transformation from c-GST to h-GST, the TBC at the h-GST/W interface considerably decreases. As a result, we observe that the effective thermal conductivity for the 40 nm GST film deviates from the 160 nm GST in the hexagonal phase. Above 300 °C, we no longer measure the intrinsic thermal conductivity of the h-GST layer but, instead, a convolution of the h-GST thermal conductivity and the h-GST/W thermal boundary conductance. The effective thermal conductivity for the 40 nm sample plateaus near ~0.8 W m$^{-1}$ K$^{-1}$, almost a factor of two lower than the thermal conductivity measured for 160 nm h-GST. This difference is even more pronounced when the samples are cooled down to room temperature where the thermal conductivity for 40 and 160 nm films are ~0.5 and ~1.3 W m$^{-1}$ K$^{-1}$, respectively. In order to ensure the observed reduction in the effective thermal conductivity is not due to any microstructural changes in the film, we present extensive TEM with in situ heating to compare the quality of the crystals for both thicknesses. Although defects, such as stacking faults, occur in the hexagonal phase as shown in Figs. 3e, f, we did not identify any significant microstructural anomalies between the two cases that explain such a significant reduction in the 40-nm-thick GST film. The TEM results imply that the intrinsic thermal conductivity in both cases remains unaltered and, therefore, the observed discrepancy must be related to extrinsic effects such as TBC. This finding indicates that, contrary to what is generally assumed, total thermal transport does not necessarily increase with the increase in thermal conductivity of GST.

To further support our hypothesis regarding the effect of TBC on thermal transport in h-GST, we measure the total thermal conductance across the Ru/W/GST/W/Si film stack for a 20-nm-thick GST layer. For this thickness regime, the Kapitza length, defined as the thermal conductivity divided by the TBC and represents the thickness of a material in which TBCs can influence the overall thermal transport of a system, is comparable to the thickness of the film and, as a result, the effect of interfaces in our measurements are more pronounced compared to the 40 nm film. For this thickness, the thermal conductances as a function of temperature are depicted in Fig. 3b in solid diamonds, which follow the same trend observed in effective thermal conductivity of 40 nm film with a more pronounced drop at the transition from c-GST to h-GST. This is clear evidence for the opposite trend of TBC to that of the thermal conductivity for c-GST to h-GST transition. In order to compare the TBC for c-GST vs. h-GST, we take another 20-nm-thick a-GST sample and heat it up to 320 °C where the GST film becomes fully cubic. By cooling the sample down to room temperature, we measure the thermal conductance in the cubic phase as a function of temperature. Once the thermal conductance over the temperature range of interest was measured, the same sample is again heated up to 400 °C to transform the c-GST into hexagonal phase and its thermal conductance was remeasured upon cooling (hollow diamonds). As shown in Fig. 3b, we obtain a higher thermal conductance in c-GST than that of the h-GST phase across the entire temperature range. This is contrary to expectations as the thermal conductivity in h-GST is nearly two times higher than the c-GST, however, due to relatively poor thermal transport at the interfaces, a lower thermal conductance is measured. Based on the results presented here, we conclude that the TBC between h-GST and W is lower than that of the c-GST and W.

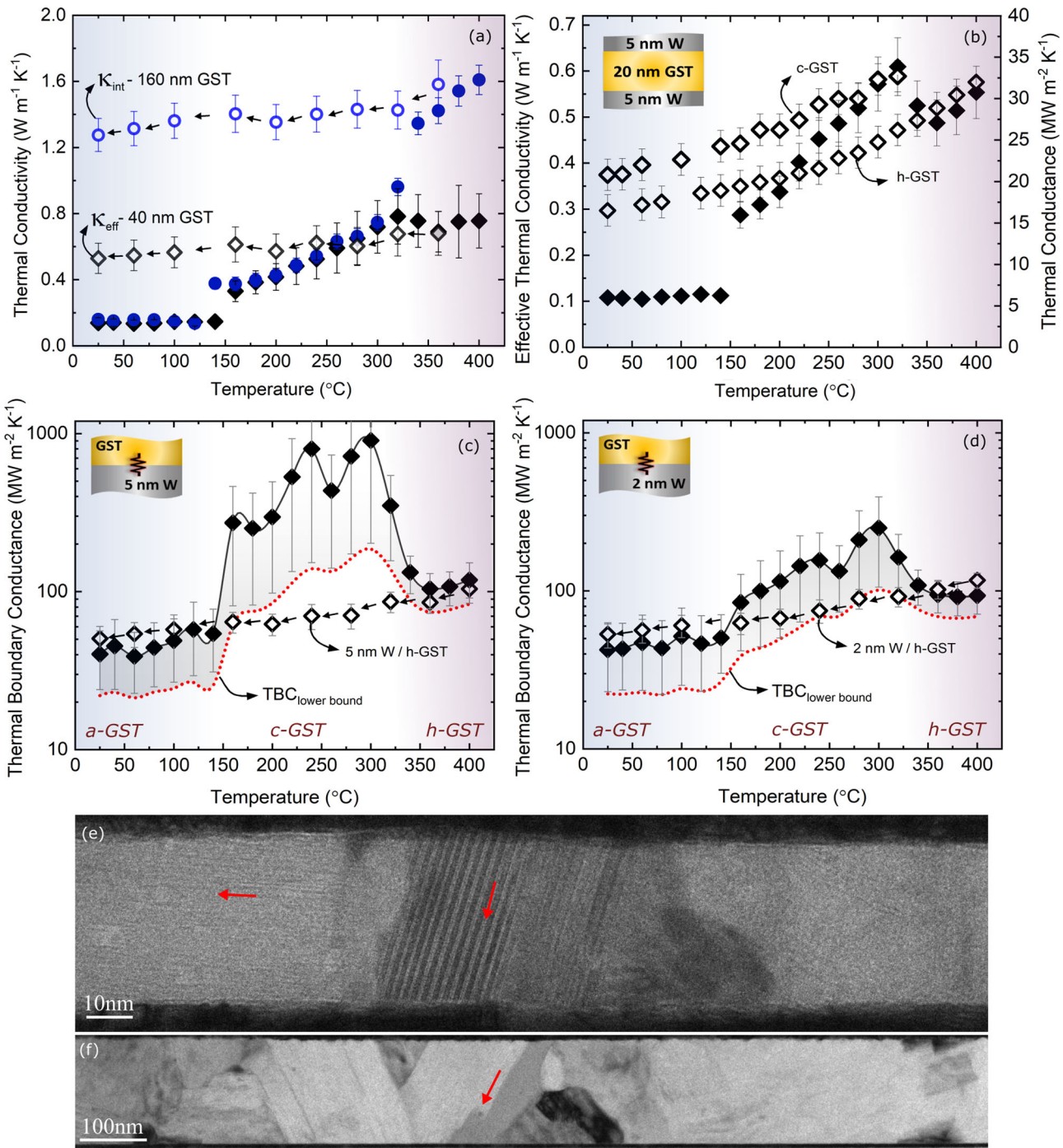

**Fig. 3 Thermal properties of GST thin films and interfaces across different phases. a** Thermal conductivity of GST layer sandwiched between 5 nm W spacers for 40 nm (diamonds) and 160 nm (circles) GST films. The solid symbols correspond to the thermal conductivity of GST as it transitions through different phases upon heating and hollow symbols correspond to the thermal conductivity of h-GST upon cooling. **b** Effective thermal conductivity for 20-nm-thick GST film as a function of temperature across different phases of GST upon heating and cooling when annealed to 320 and 400 °C. **c, d** Thermal boundary conductance for as-deposited GST upon heating and h-GST upon cooling with 5 and 2 nm W spacer, respectively. The error bars are calculated based on 7% uncertainty in the GST film thickness. **e, f** Bright-field images of 40 and 160 nm GST films at 400 °C.

In this respect, due to the significant impact of TBC on thermal transport of thin film GST, it is important to study how it changes across various phases. Figures 3c, d show the TBC between GST and two different thicknesses of W. The TBC in h-GST is significantly suppressed compared to c-GST. A similar reduction in TBC has been experimentally and theoretically observed at GST/metal interfaces, which were attributed to the formation of a 2-nm interfacial layer[49] and increased electron–phonon contribution to

the interfacial resistance[50]. According to our TEM images, we do not observe an additional interfacial layer after cubic to hexagonal phase transition. Considering the reduction of structural disorder upon annealing GST to higher temperatures[47,48], we use a simplistic model via molecular dynamics simulations and demonstrate that a change in atomic-scale disorder at the interface from c-GST to h-GST can, in fact, be another possible reason behind the suppression of thermal transport. Disorder and defects at interfaces

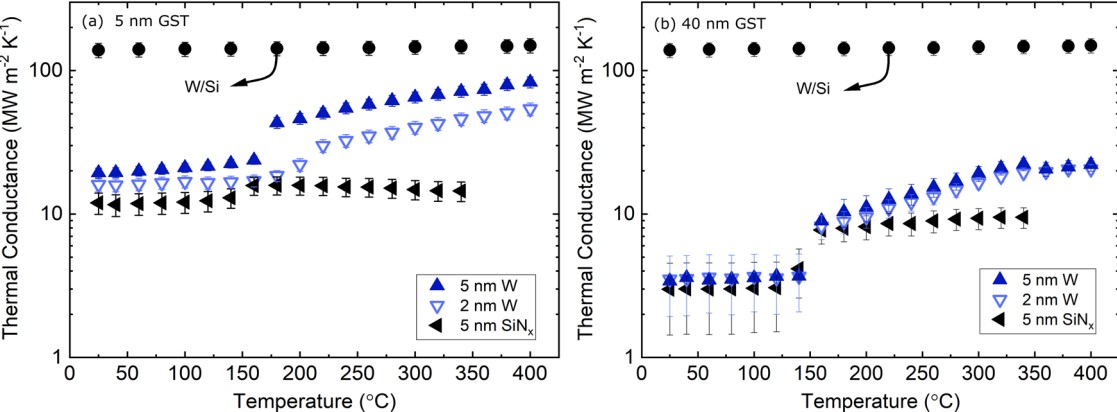

**Fig. 4 Thermal conductance behavior of thin GST films in the ballistic and diffusive regime.** Thermal conductance for **a** 5- and **b** 40-nm-thick GST in contact with different spacers. The error bars are calculated based on 7% uncertainty in the GST film thickness. Experimental data are not available for SiN$_x$ above 340 °C due to film delamination.

are well known to influence the TBC, and have in fact been computationally and experimentally shown to enhance TBC[51–54]. To this end, our molecular dynamics simulations suggest that interfacial disorder plays a stronger role in the reduction of TBC than changes in the GST crystal structure and phonon density of states. We note that in our molecular dynamics simulations, we are using Lennard–Jones potential that are not developed to predict the thermal properties of W or GST. However, the simplicity of these potentials allows us to assess our hypotheses to general classes of materials, thus providing means to broadly study our posits of the origin of reduction in TBC across the cubic to hexagonal phase transition (see Supplementary Note 3). In Figs. 3c, d, the dotted lines represent the minimum limit to the TBC by assuming the worst-case scenario for the effective parameters based on 10% uncertainty, which in many cases is far higher than measured uncertainty. Lack of sensitivity in the amorphous and cubic phases causes the reported range of TBC (best fit to minimum limit) to be quite broad, but as the GST transitions to hexagonal and gains more sensitivity to TBC, this range contracts. The hollow diamonds in Figs. 3c, d show the TBC for h-GST, which decreases by almost a factor of two as the sample is cooled down from 400 °C to room temperature. Figures 3c, d again demonstrate that the TBC for 5 nm W is higher than that of 2 nm W, especially in the crystalline phase where the effect of TBC is more pronounced.

Figures 4a, b show the thermal conductance across Ru/W/GST/W/Si as a function of temperature for 5- and 40-nm-thick GST film with different spacers (2 nm W, 5 nm W, and 5 nm SiN$_x$). The thermal conductance for a 10 nm W control is also plotted to clarify that the TBC at Ru/W and W/Si interfaces are relatively constant and sufficiently large compared to GST intrinsic thermal conductivity and GST/W interface. In Fig. 4a, we observe a linear trend for thermal conductance of 5 nm GST film as a function of temperature after the crystallization onset (>150 °C) for W spacer. This is in contrast with the trend observed in Fig. 4b for 40-nm-thick GST where the thermal conductance plateaus above 300 °C. In a fully diffusive thermal transport regime, the effect of reduced TBC in h-GST must be even more noticeable for 5 nm GST where the effect of intrinsic thermal conductivity is minimum. To explain this, it has been shown that as the thickness of the GST layer decreases to ultra-thin, the onset of crystallization increases to higher temperatures[55]. As a result, it is tempting to attribute this increase to a crystallization lag where at 300-400 °C range the film is gradually transitioning to h-GST. To assess this hypothesis, we can predict the total thermal conductance of

the Ru/5 nm W/5 nm GST/5 nm W/Si stack at different phases using thermal conductivity and TBC measured in the previous section. Using a series resistors model, we calculate the stack total thermal conductance at 400 °C to be $43 \pm 5$ MW m$^{-2}$ K$^{-1}$, which is almost a factor 2 lower than the measured value of $83 \pm 7$ MW m$^{-2}$ K$^{-1}$. It is noteworthy to mention that the TBC for GST/W and W/GST interfaces alone is $59 \pm 7$ MW m$^{-2}$ K$^{-1}$, which is significantly lower than the measured thermal conductance for the entire stack. The fact that we measure almost a factor of two higher thermal conductance for h-GST at 400 °C cannot be explained within the diffusive thermal transport limit. On the other hand, it has been shown that in bulk tungsten the average electron mean free paths before scattering with phonons at room temperature can be as long as 19.1 nm[56]. Additionally, due to tungsten high lattice thermal conductivity (~46 W m$^{-1}$ K$^{-1}$)[56], it has phonons with long mean free paths relative to other metals[57]. From this, we conservatively estimate the phonon mean free path in tungsten to be on the order of $\lambda = 3k_p/Cv = (3 \times 46)/(2.58 \times 10^6 \times 5174) = 10.3$ nm. Since the thickness of 5 nm GST is within the range of the phonons' and electrons' mean free paths, we attribute this enhancement in thermal conductance to ballistic transport of energy carriers emitted from the top W spacer to the bottom W spacer.

We further study this hypothesis by measuring the thermal conductance of a similar multilayer system in which we replace the W spacer with amorphous SiN$_x$. The SiN$_x$ spacer is widely used as a dielectric in electronic devices due to its high electrical and thermal resistivity[28]. As a result, we expect the contribution of phonons and electrons to thermal conductance in SiN$_x$ to be negligible compared to that of W spacer. As shown in Fig. 4a the thermal conductance across the Ru/SiN$_x$/5 nm GST/SiN$_x$/Si stack is relatively constant across all temperatures, which gives further credence to our hypothesis regarding ballistic electron/phonon transport leading to increased thermal conductance in the 5 nm GST films between W contacts. In this case, no significant enhancement is observed in thermal conductance after the crystallization temperature. This suggests that, due to the absence of long-wavelength electrons and phonons in amorphous SiN$_x$, there is no ballistic transportation from the top SiN$_x$ to the bottom SiN$_x$ layer.

**Sound speed measurements.** In order to gain further insight into the thermal properties of GST layers below 40 nm, picosecond ultrasonic measurements are used to measure the sound speed of the GST thin films. In these measurements, the absorption of the

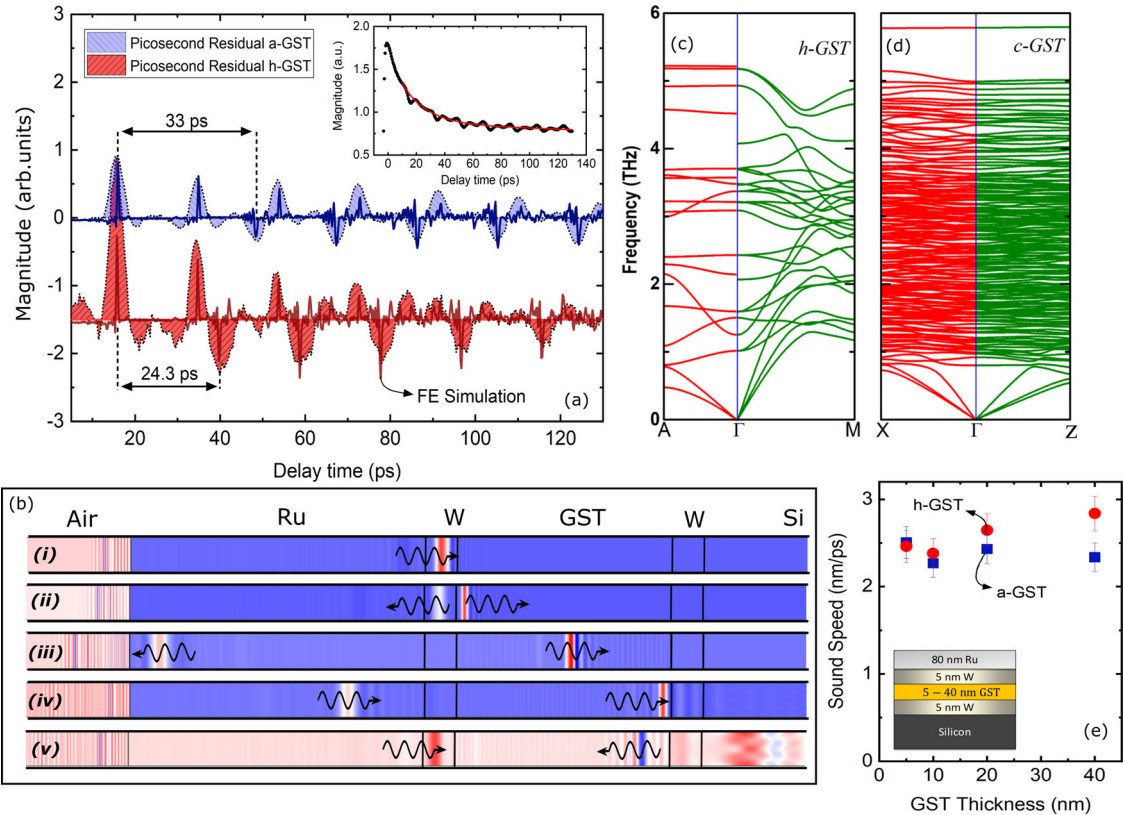

**Fig. 5 Sound speed measurements for thin GST films. a** Picosecond ultrasonic measurements for 40 nm of a-GST (blue) and 400 °C annealed h-GST (red). The "humps" and "troughs" corresponds to the reflection of strain waves off of W/GST and GST/W interfaces, respectively. The solid lines correspond to finite-element simulation of strain wave propagation across different layers. **b** A representation of strain wave and how it propagates and reflects off of various interfaces. **c**, **d** Calculated phonon dispersions for h-GST and c-GST. **e** Sound speed measurements for different thicknesses of a-GST and h-GST. The error bars are calculated based on 10% uncertainty for the GST film thicknesses.

ultrashort laser pulse launches a strain wave from the sample surface due to the rapid heating. This results in qualitative "humps" and "troughs" superimposed on the TDTR thermal decay curve. The temporal spacing of these picosecond ultrasonic signals are related to the time it takes for the strain waves to travel through a material and reflect off of sub-surface interfaces as demonstrated in schematic in Fig. 5b. Thus, with knowledge of the thicknesses of each film (determined via TEM), we can estimate the sound speed. For a better interpretation of picosecond ultrasonic data (the inset in Fig. 5a), the thermal decay curve is subtracted from the best exponential fit to the experimental data and the residual is presented in Fig. 5a, where the "humps" and "troughs" in the plot are the consequence of strain waves reflection from the top and bottom interfaces of the GST layer.

By measuring the time between the upward "humps" and downward "troughs" in Fig. 5a, we obtain the time it takes for the strain waves to travel across the GST. Based on the measured thickness of the GST film, the longitudinal sound speed in 40 nm a-GST and h-GST layers is measured to be $2300 \pm 200$ and $2800 \pm 200$ m s$^{-1}$, respectively. Although, our sound speed measurement for a-GST agrees with literature, the sound speed for h-GST is below what has been reported[12,58]. To further understand the discrepancy between our sound speed measurements and previously reported value in the hexagonal phase, we perform first-principle calculations for GST in both cubic and hexagonal phases. Figures 5c, d show the phonon dispersions for h-GST and c-GST. For the cubic phase, the average group velocities for the LA and TA modes were calculated as 3531 and 1658 m s$^{-1}$, respectively. Due to the structural anisotropy, we observe strong anisotropic phonon dispersion in h-GST, giving anisotropic group

velocities along the in-plane (LA = 3828 m s$^{-1}$; TA = 2398 m s$^{-1}$) and out-of-plane directions (LA = 3502 m s$^{-1}$; TA = 2567 m s$^{-1}$). The similar sound speed between c-GST and h-GST along the out-of-plane direction is consistent with our observation from the picosecond ultrasonic measurements. However, quantitatively, higher LA group velocity in our calculation compared to the measured value is in contrast with the typical under-binding tendency of the generalized gradient approximation that increases bond length and softens the phonons leading to lower group velocities compared to that in measurements. This implies that the LA mode velocities in h-GST and c-GST thin films are lower than that in their bulk counterpart. This reduction is more prominent when the sound speed for different thicknesses are measured and plotted in Fig. 5e. As can be seen, as the thickness of GST decreases the sound speed in hexagonal phase converges to that of the amorphous phase. The reduced measured sound speed for h-GST is most likely due to the existence of amorphous regions near the interface, and we hypothesize not intrinsic to the GST. For more details on sound speed measurements refer to the Supplementary Note 4.

## Discussion

As the memory cell dimension in PCM devices shrink and progress towards superlattice structures, it is essential to account for the parameters that are not conventionally considered in thick regime such as interfacial thermal resistance and ballistic thermal transport. In superlattice PCMs, due to existence of several interfaces in a single cell, engineering the interfacial resistance can substantially improve the performance of the device. For superlattice structures, the TBR at GeTe/Sb$_2$Te$_3$ interface is reported[59] to be around 3.4

$m^2$ K GW$^{-1}$. Our work, in addition to reporting a significantly higher TBR between h-GST and W, ~10 m$^2$ K GW$^{-1}$, demonstrates how judiciously engineering the interface between GST and its adjacent material can reduce the reset current. To demonstrate the effect of TBR on thermal transport, we showed that for a 20-nm-thick GST film the effective thermal conductivity can be reduced by a factor of 4 due to the increased interfacial resistance in h-GST. Our work shows that interfacial resistance is only effective in reducing thermal transport when the GST thickness is less than 40 nm. On the other hand, there is a limitation on reducing the thickness of the GST layer before the thermal transport transitions into a ballistic regime. According to our results, as the thickness of GST reaches ~5 nm, ballistic transport of phonons/electrons from the top W electrode to the bottom electrode increases the thermal transport by almost a factor of two. To prevent this ballistic transport effect, it is important to choose interlayers that have carriers with short mean free paths. In our previous work[60] we demonstrated that materials such as carbon nitride with short mean free path energy carriers can serve as a better electrode than tungsten when the device dimension reaches below 10 nm. For the specific layer configuration studied here, W/GST/W, the GST thickness at which electrode engineering has the biggest impact in efficiency optimization, is approximately 20 nm. For thinner GST thicknesses, ballistic thermal transport limits thermal confinement and at larger thicknesses, bulk properties of the GST will play a larger role and as a result the effect of TBR between GST and the electrode diminishes.

Earlier, we demonstrated that by reducing the W thickness from 5 to 2 nm, thermal conductance can be moderately suppressed. In order to demonstrate the effect of W layer thickness on PCM device performance, we use computational models for a PCM device in confined cell geometry. For this, a 35-nm-thick GST unit is sandwiched between identical W layers (2 or 5 nm), and connected to TaN electrodes. The cell geometry is a cylinder confined by dielectric materials; we repeat our simulation for two different cell dimensions with lateral size of 20 and 120 nm diameter in order to compare thermal transport in small and large devices. The simulations are carried out by using finite-element simulation package COMSOL Multiphysics. Table 1 summarizes the step-by-step simulation process as we progressively add measured parameters into the simulation. Our simulations demonstrate that thinning the W layer from 5 to 2 nm, taking 35% reduction in thermal conductance into account, leads to reset current ($I_{reset}$) drop from 133 to 127 μA for the 20 nm device and from 3.07 to 2.88 mA for the 120 nm device, corresponding to 4.5% and 6.2% reduction in reset current, respectively. Although manipulating W thickness leads to a modest reduction in the reset current, it should be noted that this is achieved through practical changes in an interface that is not typically optimized for its thermal properties. Further optimization along these lines

could lead to larger improvements. In order to demonstrate this, we extend our simulations to account for a range of TBR between the phase change unit and the adjacent electrode. It is expected that the TBR between GST and most materials to fall in the range of 1–100 m$^2$ K GW$^{-1}$ (ref. [61]). The result of our simulations for reset current as a function TBR between PCM/electrode and the cell configuration for a 120 nm confined cell are presented in Figs. 6a, b, respectively. Our predictions suggest that the reset current can be reduced up to ~40% and ~50% depending on the device lateral size if the TBR changes from 1 to 100 m$^2$ K GW$^{-1}$. In superlattice structures where there are multiple interfaces, the reset current can be even further reduced. Boniardi et al.[62] observed nearly 60% reduction in set and reset current for (GeTe/Sb$_2$Te$_3$)/Sb$_2$Te$_3$ superlattice compared to bulk GST, which they attributed to increased thermal resistance in the superlattices from the period interfaces as compared to the GST. Our results highlight the importance of interfacial engineering on thermal confinement of PCM memory cells.

In summary, we reported on the thermal properties of GST for thicknesses below 40 nm and compared the results against the thick film regime (160 nm). We demonstrated that as the length scale of PCM cells decrease to the dimensions of the order of carriers' mean free paths, the mechanism of heat transport drastically differs from its bulk. In addition, our results demonstrate that as the GST transition from one crystallographic phase to another, the interfacial resistance changes. The TBR for a-GST/W, c-GST/W, and h-GST/W interfaces are measured to be approximately 25 ± 5, 3 ± 1.5, 10 ± 2 m$^2$ K GW$^{-1}$, respectively. Our molecular dynamics simulations results suggest that a change in phase from cubic to hexagonal does not significantly alter the TBC. However, structural disorder at the interface plays an important role in the reduction of TBC from the cubic to the hexagonal phase. Overall, the interfacial resistance for a 20-nm-thick GST film results in a factor of 4 reduction in the effective thermal conductivity from ~1.3 to ~0.3 W m$^{-1}$ K$^{-1}$ at room temperature. Our work illustrated that the TBR can be employed to substantially suppress heat transport in phase change units. We use simulations to elucidate the effect of TBR on the reset current for two different cell sizes. According to these simulations, the TBR can lead up to 40% and 50% reduction in reset current. The results presented in this work improve our knowledge of thermal transport mechanism in ultra-thin phase change units and enable us to design PCM devices with superior performance.

## Methods

TDTR is a non-contact optical measurement technique that utilizes a modulated pump beam to create an oscillatory temperature rise at the surface of the sample, and a probe beam to measure the changes in the thermoreflectance of the surface due to the thermal excitation. In our two-tint TDTR configuration, the output of an 80 MHz Ti:sapphire femtosecond pulsed laser with a center wavelength of 808 nm is split into two paths, a pump and a probe. The pump beam is passed through an electro-optical modulator and modulated at a frequency of 8.4 MHz, while the probe beam is directed through a mechanical delay stage in order to induce a time-dependent signal relative to the arrival of pump pulses. The beams are combined by passing the probe through a high-pass filter and then both beams are directed through a ×10 objective resulting in spot radius of 22 and 11 μm for the pump and probe at the sample surface, respectively. The uncertainty for room temperature measurements are determined by considering 7% change in the thickness of the Ru transducer. For calculating the uncertainty in thermal conductivity as a function of temperature for thin film samples, since the thickness of GST film alters across different temperature ranges, we assume 7% change in the thickness of GST instead of the transducer.

For ab initio density functional calculations, we considered two special quasirandom structures with 45 atoms (10 Ge, 10 Sb, and 25 Te atoms) to represent the c-GST and a-GST. The a-GST structure was obtained from a rapid quench after molecular dynamics simulations using GGA-PBE exchange-correlation functional form as implemented in VASP[63,64]. The details of the structures can be found elsewhere[65]. For h-GST, we used the structure given by Kooi et al.[66] as it gives a more stable structure[58] compared to that given by Matsunaga et al.[67]. The harmonic interatomic force constants (IFCs) were calculated using VASP-phonopy[68]

**Table 1 The impact of parameters such as thermal conductivity ($k_{GST}$), thermal boundary resistance between GST and dielectric (TBR$_{GST/d}$) or GST and tungsten (TBR$_{GST/W}$), tungsten electrode thickness ($d_W$) extracted from the empirical measurements on the reset current ($I_{reset}$) for devices with lateral size of 20 and 120 nm.**

| $k_{GST}$ (W m$^{-1}$ K$^{-1}$) | 0.8 | $k$ ($T$) | $k$ ($T$) | $k$ ($T$) | $k$ ($T$) |
|---|---|---|---|---|---|
| TBR$_{GST/d}$ (GW$^{-1}$ m$^2$ K) | 10 | 10 | 41 | 41 | 41 |
| TBR$_{GST/W}$ (GW$^{-1}$ m$^2$ K) | 10 | 10 | 10 | TBR(T) | TBR(T) |
| $d_W$ (nm) | 5 | 5 | 5 | 5 | 2 |
| $I_{reset}$ (μA) 20 nm device | 183 | 186 | 137 | 133 | 127 |
| $I_{reset}$ (μA) 120 nm device | 3.35 | 3.40 | 3.09 | 3.07 | 2.88 |

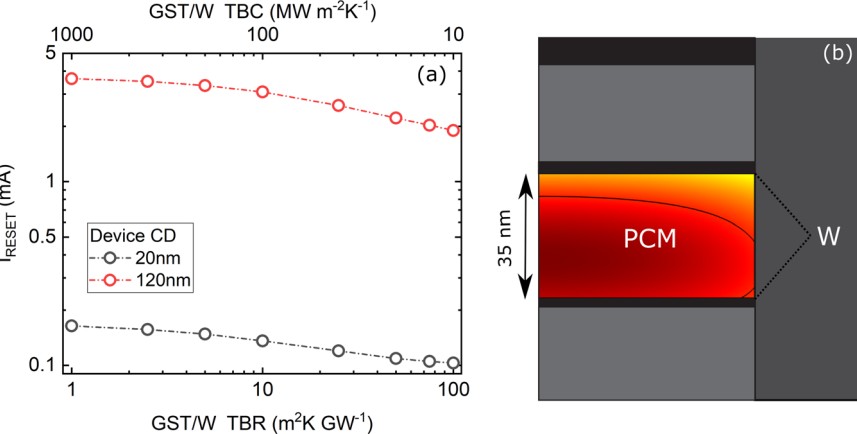

**Fig. 6 The effect of thermal boundary resistance on reset current for a confined memory cell geometry. a** Simulation results for the reset current as a function of thermal boundary resistance (TBR) between GST and W for two different device lateral sizes. **b** Schematic for the PCM configuration and its corresponding temperature gradient.

interface with a $2 \times 2 \times 2$ supercell where an energy cutoff of 500 eV was used. The calculated phonon dispersions for h-GST and c-GST are given in Figs. 5c, d. However, the calculated IFCs at 0 K resulted in imaginary modes for a-GST indicating that harmonic IFCs are not sufficient to describe the lattice dynamics of a-GST.

The finite-element simulations of the strain wave propagation shown in Fig. 5 were implemented using Structural Mechanics and Acoustics module in COMSOL Multiphysics. To form a symmetric coherent wave, periodic boundary conditions were used for the top and bottom boundary along the $Y$-axis and low reflecting boundary condition on both ends along the $X$-axis. The material properties input for these simulations are density, Young's modulus, and Poisson's ratio. The density for the amorphous and crystalline state is assumed to be 5870 and 6200 kg m$^{-3}$. The Young's modulus is obtained from our empirical sound speed measurement where we calculate 32 GPa for a-GST and 50 GPa for h-GST. The thickness of the layers are chosen to replicate the experimental values, i.e. for Ru, W, GST, and Si the thicknesses are 80, 5, 40 nm, and semi-infinite, respectively. To create the strain waves, a short displacement pulse (half-sine) is applied to the surface of the Ru and the resulting echoes, generated due to the reflection of strain waves off of different interfaces, are probed at the Ru surface in temporal resolution.

**Reporting summary**. Further information on research design is available in the Nature Research Reporting Summary linked to this article.

## Data availability
The data that support the findings of this study are available from the corresponding author upon reasonable request.

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

## Acknowledgements

We appreciate support from Western Digital Technologies, Inc. This manuscript is based upon work supported by the Air Force Office of Scientific Research under Award No. FA9550-18-1-0352. S.M. acknowledges support from NRC Research Associateship.

## Author contributions

K.A., J.N., J.T.G., M.K.G., and P.E.H designed the experiment. J.N. and J.C.R. made the samples. E.R.H. and J.M.H. characterized the samples. K.A., J.T.G., and D.H.O. performed the experiments. K.A., D.A.S., Z.B., S.M., and A.G. performed the simulations. K.A., J.T.G., M.K.G., and P.E.H wrote the manuscript.

## Competing interests

The authors declare no competing interests.
