## [Peer Review File · Nature Communications]

REVIEWER COMMENTS

Reviewer #1 (Remarks to the Author):

The present paper discusses interface controlled thermal properties of ultra-thin chalcogenide-based phase change memory devices. Given the potential impact of these memories for storage-class memories, and in-memory computing, as well as the fundamental interest in these unconventional materials, this is a very timely topic. It is also convincing to see that the present manuscript discusses the impact of interfaces on thermal conductance, a topic which has received less attention, even though the authors show convincingly, that this is quite relevant. Hence, there are a number of arguments in favor of publishing the data presented here in Nature Communications. At the same time, the present manuscript also contains a number of statements, which are either questionable or should be supported by more stringent arguments. These comments are listed chronologically in the following. While some of these comments have to be addressed (marked by the word (serious) in brackets), other comments merely suggest recommendations that should be considered (marked as minor).

- a) Figure 1. (c) implies that there is an epitaxial atomic arrangement across the interface. Yet, this creates a misleading impression, since the films are not even crystalline for many amorphous samples studied here and the crystalline samples are most likely polycrystalline. The aspect of texture and the nature of the atomic arrangement across the interface should be addressed in the manuscript and the misleading pattern should be removed in figure 1. (c). (minor)
- b) On p. 5, changes of thickness upon crystallization re discussed. To my knowledge such changes in GST were first discussed in J. Vac. Sci. Technol. A 20, 230 (2002). The authors should compare their thickness changes with the data presented in this reference or similar other data. (minor)
- c) The authors argue that the building blocks in GST are van der Waals bonded, but this is questionable, since the interlayer spacing of adjacent Te layers is too small to be compatible with true van der Waals bonding, as shown for Sb₂Te₃ and other materials (Advanced Materials 31, 1904316 (2019)). In addition, the defects found at the interface are seldom stacking faults, indicative for a pronounced coupling which is not in line with van der Waals bonding (Advanced Functional Materials 29, 1902332 (2019)). This important is an imporatr conclusion, since it is important in order to explain the unconventional boundary resistance which is largest when the ordered hexagonal phase is formed. (minor)
- d) The authors argue that the thermal conductivity increases by a factor of about 10 upon the transition from the amorphous to the hexagonal crystalline phase. While this is plausible, it seems important and noteworthy, it would be highly desirable to separate the contribution of electrons and phonons to the thermal conductivity. Such data are available in the literature for GST materials. (serious)
- e) The overall reduction in I_{reset} is disappointingly modest, i.e. about 5%. The authors should explain, why the improvement is so small. (serious)
- f) It would also be highly desirable if the authors could provide a suggestion why the interface has such a pronounced impact for the hexagonal crystalline phase, but not the cubic crystalline phase. Possibly a discussion of the thermoelectric performance of nano-phase separated materials could provide a potential model to explain these data (Advanced Functional Materials 31, 1910039 (2020)). (desirable)

Reviewer #2 (Remarks to the Author):

The manuscript by Aryana et al. reports the importance of thermal boundary conductance between phase change materials and metal electrodes in phase change memory. They found by the thermal conductivity measurements that an optimum GST thickness exists based on a balance between thermal conductivity and thermal boundary conductance. Moreover, they found the phase dependence of thermal boundary conductance. I think the manuscript reads well and has useful information. However, I have some questions and comments which should be addressed.

1. Page 5, line 34, Gb-Se-Te would be typo, and should be Ge-Sb-Te.

2. The abbreviation of TBC is not explained in the main text, but only Keywords. It would be helpful to be mentioned when it appears for the first time.

3. Figure 3 (a) can mislead readers. It is explained that "the solid symbols correspond to the thermal conductivity of a-GST". However, the phase is not amorphous at all above 150°C due to crystallization. This is the in situ measurements of temperature dependence of thermal conductivity from RT to 400°C and back to RT. The same symbol can be used in the entire measurement temperature range, and only the purpose to use different symbols may be to describe different thickness samples. Something like this paper would be OK (Appl. Phys. Lett. 89, 151904 (2006)). Too many different symbols are hard to read. Temperature-dependent phases (a, c, or h) can be shown at the bottom of figure like (c). Or the authors can remove the x-axes of (a) and (b) to show the x-axes of (a) and (c) are the same.

4. In the present work, the hexagonal phase is polycrystalline. On the other hand, h-GST has a layered structure where van der Waals gaps separate each building block. Recently, superlattice-type phase change memory (iPCM) has gained attention (Nat. Nanotechnol. 6, 501 (2011)). In this case, highly oriented GeTe/Sb₂Te₃ superlattice films are obtained and they have van der Waals gaps always parallel to the substrate surface suggesting an anisotropic thermal properties of materials between x-y plane and z-axis (Phys. Status Solidi B, 252, 2151 (2015)). Indeed, the effects of thermal-based transition in superlattice phase change memory was recently reported (Phys. Status Solidi RPL, 13, 1800634 (2019)). Could the authors comment how relevant the present study is to "ordered" phase change memory?

5. From page 12 to 13, there are many "Fig.1", but probably they are "Fig. 5". Please check them.

6. The authors stated that reduction of W thickness takes no additional cost. On the other hand, industries are happy to add additional layers between GST and W even though it may cost more, if the insertion of the heat protective layer could significantly reduce the power consumption. Could the authors comment how can they further reduce I_{reset} by introducing the barrier layer (oxides or nitrides usually) in addition to the optimization of TBR and W thickness?

7. It is not clear to me whether the authors suggested to use the hexagonal phase instead of cubic phase as a SET phase. As the authors reported, the TBC could play an important role during the phase change process, but not only the thermal effects, but we should also take into account Joule heating. Since the hexagonal phase has much lower electrical resistivity than cubic, the hexagonal phase requires much current resulting in higher power consumption. Probably this is a trade-off, but should be considered.

Reviewer #3 (Remarks to the Author):

In their paper "Interface controlled thermal properties ..." Aryana et al. report the results of detailed measurements of the thermal properties of different PCM/metal and PCM/insulator heterostructures in a range of thicknesses relevant for PCM-based memory devices. The main result of this work – in my view – is the finding that the overall thermal properties of such a device-like structure can be significantly tuned through the thicknesses of the different constituents as well as the phase of the PCM-layer. It is in particular emphasized that the transition from cubic GST to hexagonal GST at elevated temperature reduces the thermal boundary conductance considerably. This might lead to more energy efficient devices since the thermal energy required to switch remains confined in the PCM-layer.

While I have no doubt that the presented results are technically sound (the group of Patrick Hopkins is well established in the application of thermal-reflectance techniques for the determination of the thermal properties of materials), I do not see that this work has sufficient impact to warrant publication in Nature Communications. I find this work much more appropriate for a journal like the Journal of Applied Physics. The manuscript is not easy to read with all its details and frequent references to the supporting material. Most important, while the reduced TBC of the h-GST/W-interface is remarkable, no real physical explanation is provided, e.g. identification of the underlying changes of the interface properties at a microscopic level. Moreover, the reduced TBC might improve the thermal efficiency by increasing the energy confinement, a much higher temperature is needed to transform to this phase, which reduces the thermal efficiency. This aspect is not discussed at all. In summary, I can not recommend publication of this manuscript in Nature Communications since in my opinion this work lacks impact, is too technical and provides no real physical insight into the observed phenomena (in particular the reduced TBC of the h-GST/W-interface).

Reviewer 1

The present paper discusses interface controlled thermal properties of ultra-thin chalcogenide-based phase change memory devices. Given the potential impact of these memories for storage-class memories, and in-memory computing, as well as the fundamental interest in these unconventional materials, this is a very timely topic. It is also convincing to see that the present manuscript discusses the impact of interfaces on thermal conductance, a topic which has received less attention, even though the authors show convincingly, that this is quite relevant. Hence, there are a number of arguments in favor of publishing the data presented here in Nature Communications. At the same time, the present manuscript also contains a number of statements, which are either questionable or should be supported by more stringent arguments. These comments are listed chronologically in the following. While some of these comments have to be addressed (marked by the word (serious) in brackets), other comments merely suggest recommendations that should be considered (marked as minor).

Authors Response:

We thank the reviewer for the positive and constructive feedback and appreciate the opportunity to address the reviewer's concerns. We have addressed all the reviewer's concern in the revised version of the manuscript, as detailed below.

a) Figure 1. (c) implies that there is an epitaxial atomic arrangement across the interface. Yet, this creates a misleading impression, since the films are not even crystalline for many amorphous samples studied here and the crystalline samples are most likely polycrystalline. The aspect of texture and the nature of the atomic arrangement across the interface should be addressed in the manuscript and the misleading pattern should be removed in figure 1. (c). (minor)

Authors Response:

We appreciate the reviewer's response and feedback. We completely agree with the reviewer's comment and removed the pattern from the schematic in Fig. 1 (c).

b) On p. 5, changes of thickness upon crystallization re discussed. To my knowledge such changes in GST were first discussed in J. Vac. Sci. Technol. A 20, 230 (2002). The authors should compare their thickness changes with the data presented in this reference or similar other data. (minor)

Authors Response:

We would like to thank the reviewer for recommending the related paper to compare our experimental measurements against. The following statement is added to the paper in the commensurate section (GST morphology at different phases), page 5:

“On average, the thickness of the GST film changes by ~5% and ~6%, at the transition from amorphous to cubic and cubic to hexagonal, respectively, which is comparable to the values of 6.5% and 8.2% reported elsewhere [1].”

[1] Njoroge *et al.* "Density changes upon crystallization of Ge₂Sb_{2.04}Te 4.74 films." Journal of Vacuum Science & Technology A: Vacuum, Surfaces, and Films 20.1 (2002): 230-233.

c) The authors argue that the building blocks in GST are van der Waals bonded, but this is questionable, since the interlayer spacing of adjacent Te layers is too small to be compatible with true van der Waals bonding, as shown for Sb₂Te₃ and other materials (Advanced Materials 31, 1904316 (2019)). In addition, the defects found at the interface are seldom stacking faults, indicative for a pronounced coupling which is not in line with van der Waals bonding (Advanced Functional Materials 29, 1902332 (2019)). This is an important conclusion, since it is important in order to explain the unconventional boundary resistance which is largest when the ordered hexagonal phase is formed. (minor)

Authors Response:

- We would like to thank the reviewer for the instructive comment. We discuss the possibilities related to the enhancement of the interfacial resistance in a subsequent question posed by the reviewer. Nonetheless, after carefully reading the suggested papers, we corrected the “van der Waals bond” to the more accurate term “metavalent bond”. Additionally, we changed the "stacking fault" to the "transitional shear fault" in the manuscript. We also added the following statement to the manuscript, page 6:

“Recent literature suggests that the weak bonding between the sesqui-chalcogenides building blocks, such as Sb₂Te₃ significantly exceeds that of a van der Waals force and therefore is more of a metavalent bond nature [1, 2].

[1] Cheng, Yudong, et al. "Understanding the Structure and Properties of Sesqui-Chalcogenides (ie, V₂VI₃ or Pn₂Ch₃ (Pn= Pnictogen, Ch= Chalcogen) Compounds) from a Bonding Perspective." *Advanced Materials* 31.43 (2019): 1904316.

[2] Mio, Antonio M., et al. "Impact of Bonding on the Stacking Defects in Layered Chalcogenides." *Advanced Functional Materials* 29.37 (2019): 1902332.

- Additionally, we added the following references to the manuscript:

[3] Wang *et al.* “Strain Tuning in Weakly Coupled Heterostructures” *Adv. Funct. Mater.* 2018, 28, 1705901.

[4] Wang *et al.* “Unconventional two-dimensional germanium dichalcogenides” *Nanoscale* 2018, 10, 73637368

d) The authors argue that the thermal conductivity increases by a factor of about 10 upon the transition from the amorphous to the hexagonal crystalline phase. While this is plausible, it seems important and noteworthy, it would be highly desirable to separate the contribution of electrons and phonons to the thermal conductivity. Such data are available in the literature for GST materials. (serious)

Authors Response:

We would like to thank the reviewer for the comment regarding the electron vs. phonon contribution in thermal conductivity of GST. Although cross plane electrical resistivity measurements for GST films are beyond the capabilities of our lab (note, the thermal conductances reported in our work are measured in the cross plane direction), in supplementary note 3 we use available electrical resistivity data in the literature to differentiate the contributions of electrons and phonons to the total thermal conductivity of GST. The following sentence is added to the main manuscript, page 7 paragraph 1:

“Although electrical resistivity measurements for GST films are beyond the scope of this paper, in supplementary note 3 we use available electrical resistivity data in the literature to differentiate the contributions of electrons and phonons to the total thermal conductivity of GST.”

A common approach to estimate the thermal conductivity due to the electron contribution is the widely used equation proposed by Wiedmann and Franz (WF):

$$k = k_p + k_e$$

$$k_e = \frac{LT}{\rho}$$

where k_p and k_e are thermal conductivities due to phonon and electron contribution, respectively, L is the Lorenz number, often assumed as the low temperature value of $2.44 \times 10^{-8} \text{ W } \Omega \text{ K}^{-2}$, T is temperature, and ρ is the electrical resistivity. Using WF approximation, Cahill and coworkers reported 70% contribution from the electrons in the total thermal conductivity of h-GST. Nonetheless, when comparing the available data for the electrical resistivity of the h-GST amongst heavily cited studies, we observe a significant difference between the reported values. The table below summarizes the measured resistivities reported by various groups and the corresponding estimated thermal conductivities using the WF law.

	Resistivity for h-GST (mΩ cm)	Measurement Temperature (°C)	Annealed Temperature (°C)	Ge ₂ Sb ₂ Te ₅ Thermal Conductivity*
Kato & Tanaka ^[1]	~3	100	580	0.3034
Nirschl & coworkers ^[2]	~2	100	350	0.4551
Lee & coworkers ^[3]	~0.84	25	300	0.8656
Siegrist & coworkers ^[4]	~0.8	100	300	1.1377
Lyeo & coworkers ^[5]	~0.585	25	400	1.2429
Bragaglia & coworkers ^[6]	~0.320	0	270	2.0816

* thermal conductivity due to electron contribution

This significant variations between the electrical resistivities in different studies could be partly due to the different deposition process, composition variation, annealing time, or different measurement techniques. In this regard, Bragaglia et al. [4] reported that the resistivity of the h-GST largely depends on the degree

of order in vacancy layers. They showed that for a single crystalline h-GST where the vacancy layers are highly ordered, the electrical resistivity could be substantially lower than reported values.

According to these studies, in h-GST, depending on the degree of disorder the thermal conductivity can largely vary. This is consistent with the observation of disorder-induced metal-insulator transition in h-GST [4]. However, as the system transitions towards more order, as well as increased electron thermal conductivity the lattice thermal conductivity is expected to increase. First principle calculations demonstrate that the lattice thermal conductivity of bulk h-GST can vary in the range of 0.87-1.67 W/mK depending on the crystal orientation [7]. Similarly, using first principle calculations, Campi et al. [8] showed that by adding various scattering terms (Sb/Ge sublattice disorder and vacancies), the lattice thermal conductivity of bulk h-GST can be adjusted to reduce from ideal value of ~1.6 W/mK to experimentally reported value of ~0.45 W/mK. Perhaps, a focused study on thermal conductivity of h-GST at the metal-insulator transition would address this question. However, such study is beyond the scope of current paper.

The following discussion regarding the electrical resistivity and the contribution of the electron in thermal conductivity is added to the supplementary material, page 12:

“Supplementary Note 3 - Electron vs. phonon contribution in thermal conductivity

An important factor in thermal transport is the contribution of electron vs. phonon in thermal conductivity across different phases of GST. Application of the Wiedmann-Franz (WF) law is a common approach that makes use of electrical resistivity for estimating the electronic contribution in thermal conductivity.

$$k = k_p + k_e$$

$$k_e = \frac{LT}{\rho}$$

where k_p and k_e are thermal conductivities due to phonon and electron contribution, respectively, L is the Lorenz number $2.44 \times 10^{-8} \text{ W } \Omega \text{ K}^{-2}$, T is temperature, and ρ is the electrical resistivity. For example, Lyeo and coworkers [18] reported negligible electronic contribution in a-GST and c-GST, while 70% contribution in h-GST based on electrical resistivity measurements. However, a survey of the data available in literature, as given in table S1, for the electrical resistivity of the h-GST reveals significant variations among reported values for the electrical resistivity of h-GST, ranging by as much as an order of magnitude. This difference among the electrical resistivities in different studies could be partially due to the different deposition process, composition variation, annealing time, or different measurement techniques. For example, Bragaglia et al. [19] reported that the resistivity of the h-GST largely depends on the degree of order in vacancy layers. They showed that for single crystalline h-GST, where the vacancy layers are highly ordered, the electrical resistivity could be substantially lower than reported values.

According to these studies, in h-GST, depending on the degree of disorder the thermal conductivity can largely vary. This is consistent with the observation of disorder-induced metal-insulator transition in h-GST [17]. However, as the system transitions towards more order, as well as increased electron thermal conductivity the lattice thermal conductivity is expected to increase. First principle calculations demonstrate that the lattice thermal conductivity of bulk h-GST can vary in the range of 0.87-1.67 W/mK depending on the crystal orientation [20]. Similarly, using first principle calculations, Campi et al. [21]

showed that by adding various scattering terms (Sb/Ge sublattice disorder and vacancies), the lattice thermal conductivity of bulk h-GST can be adjusted to reduce from ideal value of ~1.6 W/mK to experimentally reported value of ~0.45 W/mK. Perhaps, a focused study on thermal conductivity of h-GST at the metal-insulator transition would address this question. However, such study is beyond the scope of current paper.”

- [1] Kato, Takayuki, and Keiji Tanaka. "Electronic properties of amorphous and crystalline Ge₂Sb₂Te₅ films." Japanese journal of applied physics 44.10R (2005): 7340.
- [2] Nirschl, T., et al. "Write strategies for 2 and 4-bit multi-level phase-change memory." 2007 IEEE International Electron Devices Meeting. IEEE, 2007.
- [3] Lee, Jaeho, et al. "Phonon and electron transport through Ge₂Sb₂Te₅ films and interfaces bounded by metals." Applied Physics Letters 102.19 (2013): 191911.
- [4] Siegrist, T., et al. "Disorder-induced localization in crystalline phase-change materials." Nature materials 10.3 (2011): 202-208.
- [5] Lyeo, Ho-Ki, et al. "Thermal conductivity of phase-change material Ge₂Sb₂Te₅." Applied Physics Letters 89.15 (2006): 151904.
- [6] Bragaglia, Valeria, et al. "Metal-insulator transition driven by vacancy ordering in GeSbTe phase change materials." Scientific reports 6 (2016): 23843.
- [7] Mukhopadhyay, Saikat, Lucas Lindsay, and David J. Singh. "Optic phonons and anisotropic thermal conductivity in hexagonal Ge₂Sb₂Te₅." Scientific reports 6 (2016): 37076.
- [8] Campi, Davide, et al. "First-principles calculation of lattice thermal conductivity in crystalline phase change materials: GeTe, Sb₂Te₃, and Ge₂Sb₂Te₅." Physical Review B 95.2 (2017): 024311.

e) The overall reduction in Ireset is disappointingly modest, i.e. about 5%. The authors should explain, why the improvement is so small. (serious)

Authors Response:

This is a fair comment and we appreciate the ability to expound on this result and its implications. Putting aside the fact that for commercial products, even more modest no-cost improvements can be the difference between shipping or delaying a product, there are two important take-aways from the observed 5% reduction. The first is that the current reduction is achieved through practical changes in an interface that is not typically optimized for its thermal properties. Further optimization along these lines could lead to larger improvements. The second is that the number of interfaces can be increased without large increases in the overall stack thickness, providing a way to further increase the vertical thermal resistance without a detrimental thickness increase that would cause process challenges.

Here, it should be noted that our current work is a demonstration of how interfaces between PCM and its adjacent layer can in practice reduce thermal transport in the cross-plane direction in a device. Further enhancement and improvement at the interface could be the subject of other studies. Nonetheless, since the emphasis of our work is on thermal boundary resistance, we performed computational simulations to demonstrate to what extent the interfacial engineering can affect the reset current. The following figure and discussion are added to the paper, page 14 paragraph 1:

“Although manipulating W thickness leads to a modest reduction in the reset current, it should be noted that this is achieved through practical changes in an interface that is not typically optimized for its thermal properties. Further optimization along these lines could lead to larger improvements. In order to demonstrate this, we extend our simulations to account for a range of TBR between the phase change unit and the adjacent electrode. It is expected that the TBR between most materials to fall in the range of 1-100 $m^2 K GW^{-1}$ [54]. The result of our simulations for reset current as a function TBR between PCM/electrode and the cell configuration for a 120 nm confined cell is presented in Fig. 6 (a) and (b), respectively. Our predictions suggest that the reset current can be reduced up to ~40% and ~50% depending on the device lateral size if the TBR changes from 1 to 100 $m^2 K GW^{-1}$. This observation highlights the impact of interface engineering on thermal confinement in the PCM memory cells.”

Figure 1. (a) Simulation results for the reset current as a function of thermal boundary resistance (TBR) between GST/W for two different device lateral sizes, (b) Schematic for the PCM configuration and its corresponding temperature gradient.

f) It would also be highly desirable if the authors could provide a suggestion why the interface has such a pronounced impact for the hexagonal crystalline phase, but not the cubic crystalline phase. Possibly a discussion of the thermoelectric performance of nano-phase separated materials could provide a potential model to explain these data (Advanced Functional Materials 31, 1910039 (2020)). (desirable)

Authors Response:

We would like to thank the reviewer for pointing to this important discussion that must be addressed in the paper. Although a more in-depth study is required to accurately pinpoint the reasons behind enhancement of TBR at h-GST/W interface, we use this opportunity to postulate several possibilities that might explain such a behavior. The following discussion is added to the supplementary materials, page 10:

“Supplementary Note 2 - Why does interfacial resistance between GST and W change as the GST transitions from cubic to hexagonal?”

Although a more in-depth study is required to accurately pinpoint the reasons behind enhancement of TBR at h-GST/W interface, we provide several possibilities that might explain such a behavior:

First, one of the important factors in explaining the thermal transport at metal/non-metal interfaces, is the phonon density of states overlap between the materials at the interface [1]. Looking at the phonon DOS, we do not observe a significant change in the spectral overlap between c-GST and h-GST. However, in the pDOS for h-GST we observe several Van Hove singularities which are not present in the c-GST. These singularities demonstrate that the thermal transport will be dominated by certain modes of vibration whereas, the pDOS for c-GST suggests a more spectrally broad-band contribution to thermal boundary conductance. This is consistent with the phononic bandgaps observed in phonon dispersion along certain direction of Brillouin zones (see figures below).

Figure 2. (a) Phonon density of states (pDOS) for c-GST [2], h-GST [3], and W[4]. (b, c) Phonon dispersion for h-GST and c-GST.

Another possibility for the increased thermal boundary resistance at h-GST/W interface could be change in the W layer phase. It has been demonstrated that for magnetron sputtered layers below ~27 nm, β -W phase occurs [6,7]. Upon annealing at 280 °C, the W transforms into α -W phase. Although, Hao et al. [6] did not observe a change in the W phase for thicknesses below ~22 nm, here the W layer is on a soft GST layer and therefore, it might be possible that this phase transition occurs for thinner layers, and some combination of this change in interfacial structure and bonding could lead to a change in thermal boundary conductance. It has been shown that β -phase tungsten with A15 crystal structure possess lower density and lower Young's modulus than α -phase [8]. Therefore, it is possible that as-deposited softer β -W contacts form a better spectral phonon overlap with GST layer than α -W.

Another possibility for the enhancement of TBR could be variations in the bonding nature at the interface as the GST transitions from cubic to hexagonal phase. It has been shown in a number of different chalcogenide-based phase change materials such as GeTe and Sb₂Te₃, that upon transition from amorphous to crystalline, the bonding nature also transitions from covalently bonded network to a metavalent bonded (MVB) matrix [9-11]. These metavalent bonds are distinctively different from their ionic or covalent counterparts in terms of the way electrons are shared between atomic pairs. MVBs are identified as soft bonds with strong anharmonicity and therefore exhibit low thermal conductivity. It has been shown that interfaces between MVB/non-MVB result in a reduction in thermal transport [12]. It is possible that as the GST transforms towards more orders, MVB becomes the dominant bond in the h-GST and therefore reduce the thermal transport at the h-GST/W interface.”

[1] Giri, Ashutosh, Jeffrey L. Braun, and Patrick E. Hopkins. "Effect of crystalline/amorphous interfaces on thermal transport across confined thin films and superlattices." *Journal of Applied Physics* 119.23 (2016): 235305.

[2] Caravati et al. "First-principles study of crystalline and amorphous Ge₂Sb₂Te₅ and the effects of stoichiometric defects." *Journal of Physics: Condensed Matter* 21.25 (2009): 255501.

[3] Campi et al. "Electron–phonon interaction and thermal boundary resistance at the interfaces of Ge₂Sb₂Te₅ with metals and dielectrics." *Journal of Physics: Condensed Matter* 27.17 (2015): 175009.

[4] Crocombette et al. "Effect of the variation of the electronic density of states of zirconium and tungsten on their respective thermal conductivity evolution with temperature." *Journal of Physics: Condensed Matter* 27.16 (2015): 165501.

[5] Hao et al. "Beta (β) tungsten thin films: Structure, electron transport, and giant spin Hall effect." *Applied Physics Letters* 106.18 (2015): 182403.

[6] Lee et al. "Growth and characterization of α and β -phase tungsten films on various substrates." *Journal of Vacuum Science & Technology A: Vacuum, Surfaces, and Films* 34.2 (2016): 021502.

[8] Lee, Sangyeop, et al. "Resonant bonding leads to low lattice thermal conductivity." *Nature communications* 5.1 (2014): 1-8.

[9] Wuttig, Matthias, et al. "Incipient metals: functional materials with a unique bonding mechanism." *Advanced Materials* 30.51 (2018): 1803777.

[10] Raty, Jean-Yves, et al. "A quantum-mechanical map for bonding and properties in solids." *Advanced Materials* 31.3 (2019): 1806280.

[11] Rodenkirchen, Cynthia, et al. "Employing Interfaces with Metavalently Bonded Materials for Phonon Scattering and Control of the Thermal Conductivity in TAGS-x Thermoelectric Materials." *Advanced Functional Materials* 30.17 (2020): 1910039.

Reviewer #2 (Remarks to the Author):

The manuscript by Aryana et al. reports the importance of thermal boundary conductance between phase change materials and metal electrodes in phase change memory. They found by the thermal conductivity measurements that an optimum GST thickness exists based on a balance between thermal conductivity and thermal boundary conductance. Moreover, they found the phase dependence of thermal boundary conductance. I think the manuscript reads well and has useful information. However, I have some questions and comments which should be addressed.

Authors Response:

We thank the reviewer for the positive and constructive feedback and appreciate the opportunity to address the reviewer's concerns. We have addressed all the reviewer's concern in the revised version of the manuscript.

1. Page 5, line 34, Gb-Se-Te would be typo, and should be Ge-Sb-Te.

Authors Response:

We thank the reviewer for pointing out this typo in the manuscript. This has been corrected in the revised version.

2. The abbreviation of TBC is not explained in the main text, but only Keywords. It would be helpful to be mentioned when it appears for the first time.

Authors Response:

We thank the reviewer for the comment regarding the TBC abbreviation. We added the extended form of TBC in its first appearance in the manuscript.

3. Figure 3 (a) can mislead readers. It is explained that "the solid symbols correspond to the thermal conductivity of a-GST". However, the phase is not amorphous at all above 150oC due to crystallization. This is the in situ measurements of temperature dependence of thermal conductivity from RT to 400oC and back to RT. The same symbol can be used in the entire measurement temperature range, and only the purpose to use different symbols may be to describe different thickness samples. Something like this paper would be OK (Appl. Phys. Lett. 89, 151904 (2006)). Too many different symbols are hard to read. Temperature-dependent phases (a, c, or h) can be shown at the bottom of figure like (c). Or the authors can remove the x-axes of (a) and (b) to show the x-axes of (a) and (c) are the same.

Authors Response:

We thank the reviewer for this suggestion. We reduced the number of symbols to two (diamond and circles). However, we kept the hollow and solid format as is because it allows the reader to distinguish between the different heating and cooling trend. Instead, using better wording, we clarified this in the caption. Also, since the scales for Fig. 3 (a) & (b) are different we did not remove the Y-axis for the figures on the right.

“Figure 3.(a) Thermal conductivity of GST layer sandwiched between 5 nm W spacers for 40 nm (diamonds) and 160 nm (circles) GST films. The solid symbols correspond to the thermal conductivity of GST as it transitions through different phases upon heating and hollow symbols correspond to the thermal conductivity of h-GST upon cooling. (b) Total thermal conductance across Ru/5nm W/20 nm GST/5 nm W/Si as a function of temperature for different phases of GST upon heating and when annealed to 300 and 400°C. (c,d) Thermal boundary conductance for as-deposited GST upon heating and h-GST upon cooling with 5 and 2 nm W spacer, respectively. The dotted lines are the minimum limit to TBC. (e,f) Bright field images of 40 nm and 160 nm GST films at 400°C.”

4. In the present work, the hexagonal phase is polycrystalline. On the other hand, h-GST has a layered structure where van der Waals gaps separate each building block. Recently, superlattice-type phase change memory (iPCM) has gained attention (Nat. Nanotechnol. 6, 501 (2011)). In this case, highly oriented GeTe/Sb₂Te₃ superlattice films are obtained and they have van der Waals gaps always parallel to the substrate surface suggesting an anisotropic thermal properties of materials between x-y plane and z-axis (Phys. Status Solidi B, 252, 2151 (2015)). Indeed, the effects of thermal-based transition in superlattice phase change memory was recently reported (Phys. Status Solidi RPL, 13, 1800634 (2019)). Could the authors comment how relevant the present study is to “ordered” phase change memory?

Authors Response:

We thank the reviewer for their comment regarding the interfacial phase change memories (iPCM) which are receiving more and more interests. In the updated version of the paper we attempted to provide a comprehensive discussion regarding this class of memories and its ties to our work. In the manuscript we refrain from using term “iPCM” as recently a different class of superlattice structure has emerged with TiTe₂/SbTe₃ configuration. Additionally, it has been shown that the GeTe/ SbTe₃ superlattice completely transforms to ordered GST after high temperature annealing [1].

So far, the primary focus of studies regarding the iPCM including the papers suggested by reviewer are on the parameters such as electrical properties [2], deposition process [3], noise and drift [4], and retention time [5]. Recently, Shen et al. [6] demonstrated that addition of TiTe₂ multilayers between SbTe₃ reduces the thermal leakage during the reset process and decreases the reset current by 87%. Despite the fact that in layered phase change memories the interfaces are an integral part of the device, to the best of our knowledge there is no study that specifically focuses on the effect of interfaces on the thermal transport. Recently, Okabe et al. [7] measured the thermal boundary resistance between Sb₂Te₃/GeTe layers to be 3.4 m² K GW⁻¹. In our study in addition to reporting the TBR between W/GST which is nearly three times higher than that of Sb₂Te₃/GeTe, 10 m² K GW⁻¹, we propose an opportunity to take more advantage of superlattice PCM by judiciously engineering the interfaces. Our paper provides a framework from a thermal perspective for the design of this type of superlattice PCM to reduce cross-plane thermal leakage during the programming. As an instance, we demonstrate that the effective thermal conductivity of the phase change unit can be reduced by a factor of 3 due to the increased interfacial resistance. This significant reduction is without any superlattice structure. However, there is a limitation which needs to be taken into account in order to take advantage of interfacial resistance in the superlattice architecture. We showed that as the thickness of GST reaches to ~5 nm, ballistic transport of phonons/electrons from the tungsten electrodes can disrupt the effect of interfacial resistance. To tackle this, it is important to choose interlayers that have carriers with short mean free paths. In our prior work [8], we demonstrated that materials such as CN_x can serve as a better electrode than tungsten when the layers thickness reaches below 10 nm. The following discussion is added to the introduction of the paper, page 2:

“More recently, superlattice phase change memories have received a great deal of attention due to their unique capabilities, offering lower power consumption, faster programming rate, higher retention time and lower noise and drift [3, 8, 9]. Although earlier superlattice PCMs consist of GeTe/Sb₂Te₃ alternating stacks, soon it was realized that this configuration tends to intermix and transform into bulk GST at high temperature annealing [10]. Nonetheless, the idea of superlattice PCMs inspired researchers to look for alternative material configurations. Very recently, Shen et al. [8] and Ding et al. [9] showed that superlattice PCM with TiTe₂/Sb₂Te₃ configuration have superior properties compared to bulk GST.

Despite the fact that in superlattice PCMs the interface is an integral component in the performance of these devices, its effect on the overall thermal transport is heretofore unknown and unstudied.”

Also, we added the following paragraph in the discussion section of the paper, page 13:

“In superlattice PCMs, due to existence of several interfaces in a single cell, engineering the interfacial resistance can substantially improve the performance of the device. For superlattice structures, the TBR at GeTe/Sb₂Te₃ interface is reported to be around 3.4 m² K GW⁻¹. Our work, in addition to reporting a significantly higher TBR between h-GST/W, ~10 m² K GW⁻¹, demonstrates how judiciously engineering the interface between GST and its adjacent material can reduce the reset current.”

- [1] Jamo Momand, Ruining Wang, Jos E Boschker, Marcel A Verheijen, Raaella Calarco, and Bart J Kooi. Interface formation of two-and three-dimensionally bonded materials in the case of gete{sb 2 te 3 superlattices. *Nanoscale*, 7(45):19136{19143, 2015.
- [2] Boniardi, M., Boschker, J. E., Momand, J., Kooi, B. J., Redaelli, A., & Calarco, R. (2019). Evidence for thermal-based transition in super-lattice phase change memory. *physica status solidi (RRL)–Rapid Research Letters*, 13(4), 1800634.
- [3] Saito, Y., Fons, P., Kolobov, A. V., & Tominaga, J. (2015). Self-organized van der Waals epitaxy of layered chalcogenide structures. *physica status solidi (b)*, 252(10), 2151-2158.
- [4] Keyuan Ding, Jiangjing Wang, Yuxing Zhou, He Tian, Lu Lu, Riccardo Mazzarello, Chunlin Jia, Wei Zhang, Feng Rao, and Evan Ma. Phase-change heterostructure enables ultralow noise and drift for memory operation. *Science*, 366(6462):210{215, 2019.
- [5] RE Simpson, P Fons, AV Kolobov, T Fukaya, M Krbal, T Yagi, and J Tominaga. Interfacial phase-change memory. *Nature nanotechnology*, 6(8):501, 2011.
- [6] Jiabin Shen, Shilong Lv, Xin Chen, Tao Li, Sifan Zhang, Zhitang Song, and Min Zhu. Thermal barrier phase change memory. *ACS applied materials & interfaces*, 11(5):5336{5343, 2019.
- [7] Kye L Okabe, Aditya Sood, Eilam Yalon, Christopher M Neumann, Mehdi Asheghi, Eric Pop, Kenneth E Goodson, and H-S Philip Wong. Understanding the switching mechanism of interfacial phase change memory. *Journal of Applied Physics*, 125(18):184501, 2019.
- [8] Aryana, JT Gaskins, J Nag, JC Read, DH Olson, MK Grobis, and PE Hopkins. Thermal properties of carbon nitride toward use as an electrode in phase change memory devices. *Applied Physics Letters*, 116(4):043502, 2020.

5. From page 12 to 13, there are many “Fig.1”, but probably they are “Fig. 5”. Please check them.

Authors Response:

We thank the reviewer for pointing out to this error in the paper. We corrected the figures number in the updated version of the paper.

6. The authors stated that reduction of W thickness takes no additional cost. On the other hand, industries are happy to add additional layers between GST and W even though it may cost more, if the

insertion of the heat protective layer could significantly reduce the power consumption. Could the authors comment how can they further reduce I_{reset} by introducing the barrier layer (oxides or nitrides usually) in addition to the optimization of TBR and W thickness?

Authors Response:

We thank the reviewer for commenting on this matter. We agree with the reviewer as the industry would not have any problem with adding more layers if it makes the device performance better. Therefore, we removed the following sentence from the paper that might send a different message:

This was removed: "This counter-intuitive observation can be readily used in any memory cell architecture at no cost."

Additionally, in order to demonstrate the effect of interfacial resistance between any combination of material that is contact with the phase change unit, we performed simulations to show how thermal boundary resistance (TBR) affect the reset current. The TBR value between GST and most material is expected to fall in the range of 1-100 $\text{m}^2 \text{K GW}^{-1}$ [1]. According to plot shown below, if the TBR changes from 1 to 100 $\text{m}^2 \text{K GW}^{-1}$ the reset current can be reduced by 40% and 50% for device size of 20 and 120 nm, respectively. This underlines the importance of TBR in the PCM cell design. The following discussion has been added to the paper, page 14 paragraph 1:

"Although manipulating W thickness leads to a modest reduction in the reset current, it should be noted that this is achieved through practical changes in an interface that is not typically optimized for its thermal properties. Further optimization along these lines could lead to larger improvements. In order to demonstrate this, we extend our simulations to account for a range of TBR between the phase change unit and the adjacent electrode. It is expected that the TBR between most materials to fall in the range of 1-100 $\text{m}^2 \text{K GW}^{-1}$ [54]. The result of our simulations for reset current as a function TBR between PCM/electrode and the cell configuration for a 120 nm confined cell is presented in Fig. 6 (a) and (b), respectively. According to this figure, the reset current can be reduced up to ~40% and ~50% depending on the device lateral size if the reset current changes from 1 to 100 $\text{m}^2 \text{K GW}^{-1}$. This observation highlights the impact of interface engineering on thermal confinement in the PCM memory cells."

Figure 4. (a) Simulation results for the reset current as a function of thermal boundary resistance (TBR) between GST/W for two different device lateral sizes, (b) Schematic for the PCM configuration and its corresponding temperature gradient.

The following table shows previous studies and their measured TBR between GST and various materials:

Interface	TBR ($\text{m}^2 \text{K GW}^{-1}$)	Reference
a-GST/W	~21	This work
c-GST/W	~3	This work
h-GST/W	~10	This work
GST/SiN _x	~41	This work
a-GST/ZnO: SiO ₂	~15	Kim et al. [2]
c-GST/ZnO: SiO ₂	~6.5	Kim et al. [2]
c-GST/graphene/W	~44	Ahn et al. [3]
GST/CN _x	~51	Aryana et al. [4]
GST/C	~27.5	Bozorg-Grayeli et al. [5]
GST/TiN	~5-50	Bozorg-Grayeli et al. [5]
GeTe /Sb ₂ Te ₃	~3.4	Okabe et al. [6]

[1] Reifenberg, J. P., Kencke, D. L., & Goodson, K. E. (2008). The impact of thermal boundary resistance in phase-change memory devices. *IEEE Electron Device Letters*, 29(10), 1112-1114.

[2] Kim et al., Thermal boundary resistance at Ge₂Sb₂Te₅/ZnS:SiO₂ interface, *APL* (2000).

[3] Ahn et al., Energy-Efficient Phase-Change Memory with Graphene as a Thermal Barrier, *Nano Letters* (2015).

[4] Aryana et al., Thermal properties of carbon nitride toward use as an electrode in phase change memory devices, *APL* (2020)

[5] Bozorg-Grayeli et al., Thermal conductivity and boundary resistance measurements of GeSbTe and electrode materials using nanosecond thermoreflectance, *IEEE*, (2010).

[6] Kye L Okabe, Aditya Sood, Eilam Yalon, Christopher M Neumann, Mehdi Asheghi, Eric Pop, Kenneth E Goodson, and H-S Philip Wong. Understanding the switching mechanism of interfacial phase change memory. *Journal of Applied Physics*, 125(18):184501, 2019.

7. It is not clear to me whether the authors suggested to use the hexagonal phase instead of cubic phase as a SET phase. As the authors reported, the TBC could play an important role during the phase change process, but not only the thermal effects, but we should also take into account Joule heating. Since the hexagonal phase has much lower electrical resistivity than cubic, the hexagonal phase requires much current resulting in higher power consumption. Probably this is a trade-off, but should be considered.

Authors Response:

We thank the reviewer for posing this important question and agree that it is not clear whether it would be practical or beneficial to use the hcp phase as the SET phase in devices unless it is tested. Nonetheless, we do not propose that hexagonal phase in GST should be used for the SET phase since this largely depends on the design priorities (stability/switching speed/power consumption/ heat leakage, and etc.). The purpose of our work is to present evidence that the thermal transport for thin film GST differs from those

in bulk GST, and therefore, the effect of parameters such as interfacial resistance and ballistic transport needs to be taken into account. Although the change in TBR during a phase transition is an interesting observation, the main take away is in the impact of the interface on the thermal transport, which can suppress the effective thermal conductivity by a factor of three. Along the same lines, we show that the effect of the interfaces only becomes noticeable for thicknesses below ~40 nm. On the other hand, if electrodes with large mean free paths of their thermal carriers such as W are used in the memory cell, for very thin thicknesses of GST, the effect of interfacial resistances disappears and the ballistic transport regime dominates the thermal transport, which may impair the performance of the device. This information is crucial in the design of superlattice phase change memories where the thickness of each layer is below ~10 nm with several interfaces.

Since the following paragraph adds confusion and might convey inaccurate message to the reader, we removed it from the main manuscript.

This was removed: "Generally, hexagonal phase in GST, despite its stability and low electrical resistivity, is undesirable due to its high thermal conductivity which results in thermal leakage. However, we show that the thermal boundary conductance across GST/electrode interface substantially decreases from the cubic to hexagonal phase and, as a result, the effective thermal conductivity is reduced by a factor of three."

Reviewer #3 (Remarks to the Author):

In their paper “Interface controlled thermal properties ...” Aryana et al. report the results of detailed measurements of the thermal properties of different PCM/metal and PCM/insulator heterostructures in a range of thicknesses relevant for PCM-based memory devices. The main result of this work – in my view – is the finding that the overall thermal properties of such a device-like structure can be significantly tuned through the thicknesses of the different constituents as well as the phase of the PCM-layer. It is in particular emphasized that the transition from cubic GST to hexagonal GST at elevated temperature reduces the thermal boundary conductance considerably. This might lead to more energy efficient devices since the thermal energy required to switch remains confined in the PCM-layer.

While I have no doubt that the presented results are technically sound (the group of Patrick Hopkins is well established in the application of thermal-reflectance techniques for the determination of the thermal properties of materials), I do not see that this work has sufficient impact to warrant publication in *Nature Communications*. I find this work much more appropriate for a journal like the *Journal of Applied Physics*. The manuscript is not easy to read with all its details and frequent references to the supporting material. Most important, while the reduced TBC of the h-GST/W-interface is remarkable, no real physical explanation is provided, e.g. identification of the underlying changes of the interface properties at a microscopic level. Moreover, the reduced TBC might improve the thermal efficiency by increasing the energy confinement, a much higher temperature is needed to transform to this phase, which reduces the thermal efficiency. This aspect is not discussed at all.

In summary, I cannot recommend publication of this manuscript in *Nature Communications* since in my opinion this work lacks impact, is too technical and provides no real physical insight into the observed phenomena (in particular the reduced TBC of the h-GST/W-interface).

Authors Response:

We appreciate the reviewer’s honest feedback and critique. In order to reduce the technicality of the paper and make it easier to read, we moved some of the discussion regarding TBC to the supplementary information instead of main manuscript. Regarding the impact of our paper, although we gratefully respect the reviewer’s opinion that the current manuscript lacks sufficient impact on the community and is not suitable for publication in *Nature Communications*, we would like to provide several arguments to prove otherwise.

So far, to the authors best knowledge, there is no study that experimentally shows the interfacial resistance can be engineered to reduce the thermal transport in PCM devices without incorporating additional materials at the interface. Especially with recent advancements in the development of PCM with superlattice structure where several interfaces are used in the memory cell, the importance of a study that provides perspective for engineering interfacial thermal transport is crucial. In this regard, our work not only demonstrates to what extent the TBR between phase change units and the adjacent contact can reduce thermal transport, but also provides a framework for the design of phase change memory cells from the interfacial engineering perspective. For instance, to the best of our knowledge, the effect of ballistic thermal transport as the thickness of GST approaches the mean free path of carriers has never been experimentally observed in PCM. In current work, as well as providing proof for observation of

ballistic thermal transport for 5 nm GST film, we propose solutions to dampen the effect of ballistic transport for future PCM designs. The following discussion is added to the paper, page 13, paragraph 2:

“Our work shows that interfacial resistance is only effective in reducing thermal transport when the GST thicknesses is less than 40 nm. On the other hand, there is a limitation on reducing the thickness of the GST layer before the thermal transport transitions into a ballistic regime. According to our results, as the thickness of GST reaches to ~5nm, the ballistic transport of phonons/electrons from the top W electrode to the bottom electrode increases the thermal transport by almost a factor of two. To prevent the ballistic transport effect in superlattice PCMs, it is important to choose interlayers that have carriers with short mean free paths. In our previous work [52] we demonstrated that materials such as CN_x where its energy carriers have a short mean free path can serve as a better electrode than tungsten when the device dimension reaches below 10 nm. For the specific layer configuration studied here, W/GST/W, we propose the optimum thickness for designing a PCM cell to be approximately 20 nm. In this limit, one can benefit from the effect of TBR in thermal confinement and at the same time stays away from the ballistic thermal transport.”

Additionally, based on the results in our paper and partially in response to the reviewer’s point regarding the impact of our work, we implemented additional simulations to indicate our results in a more practical manner. The following figure show how much reset current changes with respect to the TBR between GST and the top and bottom electrode. According to this plot, if the TBR changes from 1 to 100 $m^2 K GW^{-1}$ the reset current can be reduced by 40% and 50% for device size of 20 and 120 nm, respectively. This underlines the importance of TBR in the PCM cell design, as a reduction in reset current by this amount would revolutionize this technology. The following discussion has been added to the paper, page 14 paragraph 1:

“Although manipulating W thickness leads to a modest reduction in the reset current, it should be noted that this is achieved through practical changes in an interface that is not typically optimized for its thermal properties. Further optimization along these lines could lead to larger improvements. In order to demonstrate this, we extend our simulations to account for a range of TBR between the phase change unit and the adjacent electrode. It is expected that the TBR between most materials to fall in the range of 1-100 $m^2 K GW^{-1}$ [reifenberg2008impact]. The result of our simulations for reset current as a function TBR between PCM/electrode and the cell configuration for a 120 nm confined cell is presented in Fig. 6 (a) and (b), respectively. According to this figure, the reset current can be reduced up to ~40% and ~50% depending on the device lateral size if the reset current changes from 1 to 100 $m^2 K GW^{-1}$. This observation highlights the impact of interface engineering on thermal confinement in the PCM memory cells.”

Figure 5. (a) Simulation results for the reset current as a function of thermal boundary resistance (TBR) between GST/W for two different device lateral sizes, (b) Schematic for the PCM configuration and its corresponding temperature gradient.

We also appreciate the reviewer’s comment that we have not provided sufficient detail into the underlying mechanisms driving the reduction in TBC across h-GST/W interfaces as compared to that across the c-GST/W interfaces. We use this opportunity to postulate several possibilities that might explain such a behavior. The following discussion is added to the supplementary materials:

“Supplementary Note 2 - Why does interfacial resistance between GST and W changes as the GST transitions from cubic to hexagonal?”

Although a more in-depth study is required to accurately pinpoint the reasons behind enhancement of TBR at h-GST/W interface, we provide several possibilities that might explain such a behavior:

First, one of the important factors in explaining the thermal transport at metal/non-metal interfaces, is the phonon density of states overlap between the materials at the interface [1]. Looking at the phonon DOS, we do not observe a significant change in the spectral overlap between c-GST and h-GST. However, in the pDOS for h-GST we observe several Van Hove singularities which are not present in the c-GST. These singularities demonstrate that the thermal transport will be dominated by certain modes of vibration whereas, the pDOS for c-GST suggests a more spectrally broad-band contribution to thermal boundary conductance. This is consistent with the phononic bandgaps observed in phonon dispersion along certain direction of Brillouin zones (see figures below).

Figure 7. (a) Phonon density of states (pDOS) for c-GST [2], h-GST [3], and W [4]. (b, c) Phonon dispersion for h-GST and c-GST.

Another possibility for the increased thermal boundary resistance at h-GST/W interface could be change in the W layer phase. It has been demonstrated that the for magnetron sputtered layers below ~ 27 nm, β -W phase occurs [6,7]. Upon annealing at 280°C , the W transforms into α -W phase. Although, Hao et al. [6] did not observe a change in the W phase for thicknesses below ~ 22 nm, here the W layer is on a soft GST layer and therefore, it might be possible that this phase transition occurs for thinner layers, and some combination of this change in interfacial structure and bonding could lead to a change in thermal boundary conductance.

Another possibility for the enhancement of TBR could be variations in the bonding nature at the interface as the GST transitions from cubic to hexagonal phase. It has been shown in a number of different chalcogenide-based phase change materials such as GeTe and Sb_2Te_3 , that upon transition from amorphous to crystalline, the bonding nature also transitions from covalently bonded network to a metavalent bonded (MVB) matrix [8-10]. These metavalent bonds are distinctively different from their ionic or covalent counterparts in terms of the way electrons are shared between atomic pairs. MVBs are identified as soft bonds with strong anharmonicity and therefore exhibit low thermal conductivity. It has been shown that interfaces between MVB/non-MVB result in a reduction in thermal transport [11]. It is possible that as the GST transforms towards more order MVB becomes the dominant bond in the h-GST and therefore reduce the thermal transport at the h-GST/W interface.”

[1] Giri, Ashutosh, Jeffrey L. Braun, and Patrick E. Hopkins. "Effect of crystalline/amorphous interfaces on thermal transport across confined thin films and superlattices." *Journal of Applied Physics* 119.23 (2016): 235305.

[2] Caravati et al. "First-principles study of crystalline and amorphous $\text{Ge}_2\text{Sb}_2\text{Te}_5$ and the effects of stoichiometric defects." *Journal of Physics: Condensed Matter* 21.25 (2009): 255501.

[3] Campi et al. "Electron–phonon interaction and thermal boundary resistance at the interfaces of $\text{Ge}_2\text{Sb}_2\text{Te}_5$ with metals and dielectrics." *Journal of Physics: Condensed Matter* 27.17 (2015): 175009.

- [4] Crocombette et al. "Effect of the variation of the electronic density of states of zirconium and tungsten on their respective thermal conductivity evolution with temperature." *Journal of Physics: Condensed Matter* 27.16 (2015): 165501.
- [5] Hao et al. "Beta (β) tungsten thin films: Structure, electron transport, and giant spin Hall effect." *Applied Physics Letters* 106.18 (2015): 182403.
- [6] Lee et al. "Growth and characterization of α and β -phase tungsten films on various substrates." *Journal of Vacuum Science & Technology A: Vacuum, Surfaces, and Films* 34.2 (2016): 021502.
- [8] Lee, Sangyeop, et al. "Resonant bonding leads to low lattice thermal conductivity." *Nature communications* 5.1 (2014): 1-8.
- [9] Wuttig, Matthias, et al. "Incipient metals: functional materials with a unique bonding mechanism." *Advanced Materials* 30.51 (2018): 1803777.
- [10] Raty, Jean-Yves, et al. "A quantum-mechanical map for bonding and properties in solids." *Advanced Materials* 31.3 (2019): 1806280.
- [11] Rodenkirchen, Cynthia, et al. "Employing Interfaces with Metavalently Bonded Materials for Phonon Scattering and Control of the Thermal Conductivity in TAGS-x Thermoelectric Materials." *Advanced Functional Materials* 30.17 (2020): 1910039.

REVIEWER COMMENTS

Reviewer #1 (Remarks to the Author):

The authors have carefully considered the questions and comments of all referees and provide detailed additional information. Their replies to all referees are conclusive and convincing and further support their arguments. Hence, I am in favor of publication.

Reviewer #2 (Remarks to the Author):

The authors addressed all questions, and I think now the manuscript should be published. One minor thing that a role of the thermal resistance in super lattice phase change memory has been published recently. (Phys. Stat. Sol. (RRL), 13, 1800634 (2019)). The authors should discuss this paper in the revised manuscript.

Reviewer #3 (Remarks to the Author):

Without any doubt the authors have improved the manuscript. However, I still cannot support its publication in Nature Communications since two of my major concerns have not been satisfactorily addressed.

This refers to the readability of the manuscript, which I think is critical for a paper addressing the broad and diverse readership of Nature Communications. While some content has been shifted to the supplementary information the manuscript is still very technical with too many details.

More important, I am not at all satisfied with the authors response to my criticism that the microscopic physical mechanism behind the pronounced change of the TBC is not seriously addressed. Discussing in a mostly qualitative manner possible explanations in the supplementary information is by far not sufficient. What in my view is required to make this work suitable for Nature Communications is a combination of the detailed experimental work with a thorough theoretical analysis that can provide a microscopic understanding of the effect. And here I do not mean the "macroscopic" heat flow calculations the authors present (e.g. Fig. 6). While it is important to demonstrate that interface effects can have an effect on the performance of PCM-devices, their mere experimental demonstration without providing a deeper physical understanding is not sufficient to create the impact I think is required for a journal such as Nature Communications.

Reviewer #1 (Remarks to the Author):

The authors have carefully considered the questions and comments of all referees and provide detailed additional information. Their replies to all referees are conclusive and convincing and further support their arguments. Hence, I am in favor of publication.

Reviewer's response:

We would like to thank the reviewer for taking the time and suggesting our manuscript for publication.

Reviewer #2 (Remarks to the Author):

The authors addressed all questions, and I think now the manuscript should be published. One minor thing that a role of the thermal resistance in super lattice phase change memory has been published recently. (Phys. Stat. Sol. (RRL), 13, 1800634 (2019)). The authors should discuss this paper in the revised manuscript.

Reviewer's response:

We would like to thank the reviewer for taking the time and suggesting our manuscript for publication. We added the following statement to the discussion section of the paper, page 14, at end of the paragraph:

“In superlattice structures where there are multiple interfaces the reset current can even be further reduced. Boniardi et al. [57] observed nearly 60% reduction in set and reset current for (GeTe/Sb₂Te₃)/Sb₂Te₃ superlattice compared to bulk GST, which they attributed to increased thermal resistances in the superlattices from the period interfaces as compared to the GST.”

Reviewer #3 (Remarks to the Author):

Without any doubt the authors have improved the manuscript. However, I still cannot support its publication in Nature Communications since two of my major concerns have not been satisfactorily addressed.

This refers to the readability of the manuscript, which I think is critical for a paper addressing the broad and diverse readership of Nature Communications. While some content has been shifted to the supplementary information the manuscript is still very technical with too many details.

More important, I am not at all satisfied with the authors response to my criticism that the microscopic physical mechanism behind the pronounced change of the TBC is not seriously addressed. Discussing in a mostly qualitative manner possible explanations in the supplementary information is by far not sufficient. What in my view is required to make this work suitable for Nature Communications is a combination of the detailed experimental work with a thorough theoretical analysis that can provide a microscopic understanding of the effect. And here I do not mean the "macroscopic" heat flow calculations the authors present (e.g. Fig. 6). While it is important to demonstrate that interface effects can have an effect on the performance of PCM-devices, their mere experimental demonstration without providing a deeper physical understanding is not sufficient to create the impact I think is required for a journal such as Nature Communications.

Reviewer's response:

We would like to thank the reviewer for the constructive comment regarding the need for a deeper quantification of the microscopic mechanisms driving the observed change in TBC as the GST undergoes a phase transition. As described below we have conducted a series of molecular dynamics simulations that elucidate a likely potential nanoscopic mechanism that gives rise to the change in TBC as the GST undergoes a crystalline phase transition; these MD simulations further support one of our hypotheses that we posited in our prior version of this manuscript.

To this point, we want to be very up front and clear to the Reviewer that given the complexity of the structure, and lack of a proper interatomic cross potential for GST/W, providing an in-depth understanding for this *specific system* at the atomic scale is only possible via a detailed study using density functional theory, which is beyond the scope of this paper. As the GST undergoes a phase transition, parallel to the changes in lattice structure, the electronic structure and the bonding between the atoms also go through a transition. Obviously, the variation of several properties in GST as it undergoes a phase transition makes it exceedingly difficult to pinpoint the exact reasons behind the observed reduction in TBC. Nonetheless, in order to address reviewer's concern, the following statements and section which is the result of our molecular dynamic simulations is added to the main and supplementary information:

Main, Page 9, Paragraph 1: "A similar reduction in TBC has previously been observed between GST and aluminum films across the cubic to hexagonal phase transition which was attributed to the inter-diffusion of GST constituents into the aluminum layer and formation of 2 nm interfacial layer [48]. However, according to our TEM images, we do not observe an additional interfacial layer after the c-GST to h-GST phase transition. Further, it has been shown that the degree of disorder in GST decreases as

the annealing temperature increases and GST transitions into hexagonal phase [46, 47]. With that in mind, we use a simplistic model via molecular dynamics simulations and demonstrate that a change in atomic-scale disorder at the interface from c-GST to h-GST can, in fact, be another possible reason behind the suppression of thermal transport. Disorder and defects at interfaces are well known to influence the TBC, and have in fact have been computationally and experimentally shown to enhance TBC [49–52]. To this end, our molecular dynamics simulations suggest that varying atomic scale interfacial disorder could explain the TBC change across the c-GST to h-GST transition, and this disorder plays a stronger role in this TBC change than the change in the crystal lattice and phonon density of states. We note that in our molecular dynamics simulations, we are using Lennard-Jones potential that are not developed to predict the thermal properties of W or GST. However, the simplicity of these potentials allows us to assess our hypotheses to general classes of materials, thus providing means to broadly study our posits of the origin of reduction in TBC across the cubic to hexagonal transition (see supplementary note 3).

Main, Page 15, Summary: “Our molecular dynamic simulations results suggest that a change in phase from cubic to hexagonal does not significantly change the thermal boundary conductance. However, structural disorder at the interface plays an important role in the reduction of TBC from the cubic to the hexagonal phase.”

Supplementary Note 2: “As the GST undergoes the cubic to hexagonal phase transition, not only does the lattice structure change, but so do the electronic structure and the bonding. Obviously, the variation of several properties in GST makes it exceedingly difficult to pinpoint the exact reasons behind the observed reduction in TBC. Nonetheless, in order to provide more insight into the role of crystal structure and interfacial disorder on the observed transition in TBC, we conduct a series of molecular dynamics simulations of the TBC across cubic and hexagonal close packed interfaces of materials that have equivalent masses to W and GST using a 6-12 Lennard-Jones (LJ) potential. Lennard-Jones is a 2-body potential; therefore, the only free parameter is the distance between the atoms, and this enables us to create different lattice structures using the same potential. This therefore allows us to study the role of crystal structure and disorder on TBC without making any assumptions regarding changes in the bonding character form the cubic to hexagonal phases. To this extent, this highlights the advantages of conducting these molecular dynamics simulations using the LJ potential. Additionally, the simplicity of these potentials allows us to assess our hypotheses to general classes of materials, thus providing means to broadly study our posits of the origin of reduction in TBC across the crystalline phase transitions.

Figure 1. Simulation set up and the location of heat baths for the molecular dynamic calculations.

Since we are using LJ potentials to describe the crystalline W and GST films, in order to avoid confusion or misrepresentation, we call the section that represents W as *type 1* and the section that represents GST as *type 2*. With that in mind, we use parameters provided by Filippova et al. [1] for solid tungsten at room temperature ($\epsilon = 1.451420$ eV and $\sigma = 2.50374$ nm). Although according to the paper, these parameters are supposed to result in a BCC lattice structure, we observe the lattice is unstable and tends to reorient to FCC structure. Nonetheless, we use this potential since our main purpose here is to investigate the effect of structural *changes* on TBC. For the atoms in type 2 (GST), we could use a LJ potential with softer bonding energy compared to that of tungsten, yet, to keep the model as simple as possible, we use the same potential across all atom types in the GST. This allows us to only survey the effect of changes in the lattice structure. Thus, the only parameters that are different between type 1 (W) and type 2 (GST) are average atomic masses and number density. The atomic mass for type 1 is similar to that of W (183.84 u) and for type 2 is the arithmetic average of $\text{Ge}_2\text{Sb}_2\text{Te}_4$ (112.4 u). The number density for W and GST in our model is calculated to be $\sim 6.6 \times 10^{28} \text{ m}^{-3}$ and $\sim 2.7 \times 10^{28} \text{ m}^{-3}$ which stays relatively constant across all phases. For computational efficiency, a cutoff distance of 5.5 Å is used. For estimating the TBC at the interface between type 1 and type 2, we use a simulation box of 300 Å length with cross section area of $50 \times 50 \text{ Å}^2$. In order to investigate the effect of disorder at the interface, we used a melt-quench technique to amorphize type 2 (GST) atoms. However, due to ordered interface of type 1, the amorphous structure nucleates near interface and turn into a thin FCC layer at the type1/type 2 interface. We refer to this nucleated region as a disordered crystalline region which shows a higher TBC as compared to our “ordered” crystalline interfaces. The summary of our calculated TBC between different lattice structure are presented in table 1:

Table 1. Thermal boundary conductance (TBC) and resistance (TBR) across the interface between different lattice structures.

Interface	TBC (MW m ⁻² K ⁻¹)	TBR (GW ⁻¹ m ² K)
ordered FCC/FCC	838	1.193
disordered FCC/FCC*	1600	0.625
ordered FCC/HCP	1045	0.957
disordered FCC/HCP*	2700	0.370

*Disordered FCC/FCC and FCC/HCP interfaces are the cases where the initial phase of the atoms in type 2 were amorphous, however, during the course of simulation nucleated near the interfaces and grown to a polycrystalline structure.

Our results suggest that a change in phase from cubic to HCP does not significantly change the thermal boundary conductance. However, structural disorder at the interface could play an important role in the reduction of TBC from the cubic to HCP phase in our measured data across the W/GST/W interfaces. This is consistent with previous computational and experimental observations regarding the effect of disorder at the interface on the enhancement of TBC [14–17]. Tian et al. [14] used a theoretical approach - atomistic Green’s function- and showed that the interface roughness in Si/Ge can increase phonon transmission compared to an ideal sharp interface. They concluded that this effect is even more pronounced if the acoustic mismatch between the materials at the interface is large, which is the case for GST and W. Several molecular dynamics simulations [15, 18] have shown that compositionally disordered interfaces show higher TBCs than sharp interfaces. In addition, Gorham et al. [16] experimentally showed that TBC can increase across ion irradiated interfaces of Al/native oxide/Si with

sufficiently high ion dose due to compositional mixing and point defect formation. With respect to these previous works on the effect of disorder at the interface supported by our MD simulations, we hypothesize that one driving factor for the reduction in TBC from cubic to hexagonal phase could be due to the reduction of disorder rather than structural phase transition.”

Regarding the reviewer’s comment about the technicality of our work, all authors have read this work in detail to make sure that the impact of this work is clear, and that we are presenting our results in such a way to appeal to all readers while not sacrificing the scientific and technical rigor of *Nature Communications*. We believe that all of the discussions provided in the manuscript are necessary for the flow and readability of our paper, and still maintain the highest bar in presenting our technical results that is established in *Nature Communications*.

Figure 2. Molecular dynamics simulation results for the system size of 50x50x300 Å. (a,b) 3D and 2D visualization of the atomic arrangement in the simulation after 6 ns for cubic/cubic/cubic structure. (c,d) 3D and 2D visualization of the atomic arrangement in the simulation after 6 ns for cubic/disordered cubic/cubic structure. The disordered cubic phase is the result of nucleation from an amorphous phase. (e) The quality of interface after 6 million timesteps for interfaces with different quality. (f) Temperature profile along the simulation box when $\Delta E = 1.5$ eV/ps is added and subtracted from the hot and cold region depicted in red and blue (b,e). We calculate the TBC to be 838 MW/m²/K and 1600 MW/m²/K for ordered fcc/fcc and disordered fcc/fcc interfaces.

Figure 3. Molecular dynamics simulation results for the system size of $50 \times 50 \times 300 \text{ \AA}$. (a,b) 3D and 2D visualization of the atomic arrangement in the simulation after 6 ns for cubic/hexagonal/cubic structure. (c,d) 3D and 2D visualization of the atomic arrangement in the simulation after 6 ns for hexagonal/disordered fcc/hexagonal structure. The disordered cubic phase is the result of nucleation from an amorphous phase. (e) The quality of interface after 6 million timesteps for interfaces with different quality. (f) Temperature profile along the simulation box when $\Delta E = 1.5 \text{ eV/ps}$ is added and subtracted from the hot and cold region depicted in red and blue (b,e). We calculate the TBC to be $1045 \text{ MW/m}^2/\text{K}$ and $2700 \text{ MW/m}^2/\text{K}$ for ordered fcc/hcp and disordered fcc/hcp interfaces.

Figure 4. Normalized density of states for type 1 and type 2 for different cases studied here.

References

- [1] Filippova, V. P., S. A. Kunavin, and M. S. Pugachev. "Calculation of the parameters of the Lennard-Jones potential for pairs of identical atoms based on the properties of solid substances." *Inorganic Materials: Applied Research* 6.1 (2015): 1-4.
- [2] Battaglia, J. L., Schick, V., Rossignol, C., Kusiak, A., Aubert, I., Lamperti, A., & Wiemer, C. (2013). Thermal resistance at Al-Ge₂Sb₂Te₅ interface. *Applied Physics Letters*, 102(18), 181907.

REVIEWERS' COMMENTS

Reviewer #1 (Remarks to the Author):

The authors have taken further measures to improve the quality of their manuscript. The revised manuscript fully meets my expectations of a publication in Nature Communications.

Reviewer #2 (Remarks to the Author):

I think that now the paper should be published as it is.